# Lossless Compression with Probabilistic Circuits

**Anji Liu**
CS Department
UCLA
liuanji@cs.ucla.edu

**Stephan Mandt**
CS Department
University of California, Irvine
mandt@uci.edu

**Guy Van den Broeck**
CS Department
UCLA
guyvdb@cs.ucla.edu

## Abstract

Despite extensive progress on image generation, common deep generative model architectures are not easily applied to lossless compression. For example, VAEs suffer from a compression cost overhead due to their latent variables. This overhead can only be partially eliminated with elaborate schemes such as bits-back coding, often resulting in poor single-sample compression rates. To overcome such problems, we establish a new class of tractable lossless compression models that permit efficient encoding and decoding: Probabilistic Circuits (PCs). These are a class of neural networks involving $|p|$ computational units that support efficient marginalization over arbitrary subsets of the $D$ feature dimensions, enabling efficient arithmetic coding. We derive efficient encoding and decoding schemes that both have time complexity $\mathcal{O}(\log(D) \cdot |p|)$, where a naive scheme would have linear costs in $D$ and $|p|$, making the approach highly scalable. Empirically, our PC-based (de)compression algorithm runs 5-40 times faster than neural compression algorithms that achieve similar bitrates. By scaling up the traditional PC structure learning pipeline, we achieve state-of-the-art results on image datasets such as MNIST. Furthermore, PCs can be naturally integrated with existing neural compression algorithms to improve the performance of these base models on natural image datasets. Our results highlight the potential impact that non-standard learning architectures may have on neural data compression.

## 1 Introduction

Thanks to their expressiveness, modern Deep Generative Models (DGMs) such as Flow-based models (Dinh et al., 2014), Variational Autoencoders (VAEs) (Kingma & Welling, 2013), and Generative Adversarial Networks (GANs) (Goodfellow et al., 2014) achieved state-of-the-art results on generative tasks such as creating high-quality samples (Vahdat & Kautz, 2020) and learning low-dimensional representation of data (Zheng & Sun, 2019). However, these successes have not been fully transferred into neural lossless compression; see (Yang et al., 2022) for a recent survey. Specifically, GANs cannot be used for lossless compression due to their inability to assign likelihoods to observations. Latent variable models such as VAEs rely on rate estimates obtained by lower-bounding the likelihood of the data, i.e., the quantity which is theoretically optimal for lossless compression; they furthermore rely on sophisticated schemes such as bits-back coding (Hinton & Van Camp, 1993) to realize these rates, oftentimes resulting in poor single-sample compression ratios (Kingma et al., 2019).

Therefore, good generative performance does not imply good compression performance for lossless compression, as the model needs to support efficient algorithms to encode and decode close to the model's theoretical rate estimate. While both Flow- and VAE-based compression algorithms (Hoogeboom et al., 2019; Kingma et al., 2019) support efficient and near-optimal compression under certain assumptions (e.g., the existence of an additional source of random bits), we show that Probabilistic Circuits (PCs) (Choi et al., 2020) are also suitable for lossless compression tasks. This class of *tractable* models has a particular structure that allows efficient marginalization of its random variables–a property that, as we show, enables efficient conditional entropy coding. Therefore, we introduce PCs as backbone models and develop (de)compression algorithms that achieve high compression ratios and high computational efficiency.

Similar to other neural compression methods, the proposed lossless compression approach operates in two main phases — (i) learn good PC models that approximate the data distribution, and (ii) compress and decompress samples $x$ with computationally efficient algorithms. The proposed lossless compression algorithm has four main contributions:

*A new class of entropy models.* This is the first paper that uses PCs for data compression. In contrast to other neural compression algorithm, we leverage recent innovations in PCs to automatically learn good model architectures from data. With customized GPU implementations and better training pipelines, we are the first to train PC models with competitive performance compared to deep learning models on datasets such as raw MNIST.

*A new coding scheme.* We developed a provably efficient (Thm. 1) lossless compression algorithm for PCs that take advantage of their ability to efficiently compute arbitrary marginal probabilities. Specifically, we first show which kinds of marginal probabilities are required for (de)compression. The proposed algorithm combines an inference algorithm that computes these marginals efficiently given a learned PC and SoTA streaming codes that use the marginals for en- and decoding.

*Competitive compression rates.* Our experiments show that on MNIST and EMNIST, the PC-based compression algorithm achieved SoTA bitrates. On more complex data such as subsampled ImageNet, we hybridize PCs with normalizing flows and show that PCs can significantly improve the bitrates of the base normalizing flow models.

*Competitive runtimes.* Our (de)compressor runs 5-40x faster compared to available implementations of neural lossless compressors with near SoTA performance on datasets such as MNIST.[1] Our open-source implementation of the PC-based (de)compression algorithm can be found at `https://github.com/Juice-jl/PressedJuice.jl`.

**Notation** We denote random variables by uppercase letters (e.g., $X$) and their assignments by lowercase letters (e.g., $x$). Analogously, we use bold uppercase (e.g., $\mathbf{X}$) and lowercase (e.g., $x$) letters to denote sets of variables and their joint assignments, respectively. The set of all possible joint assignments to variables $\mathbf{X}$ is denoted $\mathsf{val}(\mathbf{X})$.

## 2 TRACTABILITY MATTERS IN LOSSLESS COMPRESSION

The goal of lossless compression is to map every input sample to an output codeword such that (i) the original input can be reconstructed from the codeword, and (ii) the expected length of the codewords is minimized. Practical (neural) lossless compression algorithms operate in two main phases — learning and compression (Yang et al., 2022). In the learning phase, a generative model $p(\mathbf{X})$ is learned from a dataset $\mathcal{D} := \{x^{(i)}\}_{i=1}^N$. According to Shannon's source coding theorem (Shannon, 1948), the expected codeword length is lower-bounded by the negative cross-entropy between the data distribution $\mathcal{D}$ and the model distribution $p(\mathbf{X})$ (i.e., $-\mathbb{E}_{x \sim \mathcal{D}}[\log p(x)]$), rendering it a natural and widely used objective to optimize the model (Hoogeboom et al., 2019; Mentzer et al., 2019).

In the compression phase, compression algorithms take the learned model $p$ and samples $x$ as input and generate codewords whose expected length approaches the theoretical limit (i.e., the negative cross-entropy between $\mathcal{D}$ and $p$). Although there exist various close-to-optimal compression schemes (e.g., Huffman Coding (Huffman, 1952) and Arithmetic Coding (Rissanen, 1976)), a natural question to ask is *what are the requirements on the model $p$ such that compression algorithms can utilize it for encoding/decoding in a computationally efficient manner?* In this paper, we highlight the advantages of *tractable* probabilistic models for lossless compression by introducing a concrete class of models that are expressive and support efficient encoding and decoding.

To encode a sample $x$, a standard streaming code operates by sequentially encoding every symbol $x_i$ into a bitstream $b$, such that $x_i$ occupies approximately $-\log p(x_i | x_1, \ldots, x_{i-1})$ bits in $b$. As a result, the length of $b$ is approximately $-\log p(x)$. For example, Arithmetic Coding (AC) encodes the symbols $\{x_i\}_{i=1}^D$ (define $D := |\mathbf{X}|$ as the number of features) sequentially by successively refining an interval that represents the sample, starting from the initial interval $[0, 1)$. To encode $x_i$, the algorithm

---

[1]Note that there exists compression algorithms optimized particularly for speed by using simple entropy models (Townsend et al., 2019), though that also leads to worse bitrates. See Sec. 3.3 for detailed discussion.

partitions the current interval $[a, b)$ using the left and right side cumulative probability of $x_i$:

$$l_i(x_i) := p(X_i < x_i \mid x_1, \ldots, x_{i-1}), \qquad h_i(x_i) := p(X_i \leq x_i \mid x_1, \ldots, x_{i-1}). \tag{1}$$

Specifically, the algorithm updates $[a, b)$ to the following: $[a + (b-a) \cdot l_i(x_i), a + (b-a) \cdot h_i(x_i))$, which is a sub-interval of $[a, b)$. Finally, AC picks a number within the final interval that has the shortest binary representation. This number is encoded as a bitstream representing the codeword of $\boldsymbol{x}$. Upon decoding, the symbols $\{x_i\}_{i=1}^{D}$ are decoded sequentially: at iteration $i$, we decode variable $X_i$ by looking up its value $x$ such that its cumulative probability (i.e., $l_i(x)$) matches the subinterval specified by the codeword and $x_1, \ldots, x_{i-1}$ (Rissanen, 1976); the decoded symbol $x_i$ is then used to compute the following conditional probabilities (i.e., $l_j(x)$ for $j > i$). Despite implementation differences, computing the cumulative probabilities $l_i(x)$ and $h_i(x)$ are required for many other streaming codes (e.g., rANS). Therefore, for most streaming codes, the main computation cost of both the encoding and decoding process comes from calculating $l_i(x)$ and $h_i(x)$.

The main challenge for the above (de)compression algorithm is to balance the expressiveness of $p$ and the computation cost of $\{l_i(x), h_i(x)\}_{i=1}^{D}$. On the one hand, highly expressive probability models such as energy-based models (Lecun et al., 2006; Ranzato et al., 2007) can potentially achieve high compression ratios at the cost of slow runtime, which is due to the requirement of estimating the model's normalizing constant. On the other hand, models that make strong independence assumptions (e.g., n-gram, fully-factorized) are cheap to evaluate but lack the expressiveness to model complex distributions over structured data such as images.[2]

This paper explores the middle ground between the above two extremes. Specifically, we ask: *are there probabilistic models that are both expressive and permit efficient computation of the conditional probabilities in Eq.* (1)? This question can be answered in the affirmative by establishing a new class of tractable lossless compression algorithms using Probabilistic Circuits (PCs) (Choi et al., 2020), which are neural networks that can compute various probabilistic queries efficiently. In the following, we overview the empirical and theoretical results of the proposed (de)compression algorithm.

We start with theoretical findings: the proposed encoding and decoding algorithms enjoy time complexity $\mathcal{O}(\log(D) \cdot |p|)$, where $|p| \geq D$ is the PC model size. The backbone of both algorithms, formally introduced in Sec. 3, is an algorithm that computes the $2 \times D$ conditional probabilities $\{l_i(x), h_i(x)\}_{i=1}^{D}$ given any $\boldsymbol{x}$ efficiently, as justified by the following theorem.

**Theorem 1 (informal).** *Let $\boldsymbol{x}$ be a $D$-dimensional sample, and let $p$ be a PC model of size $|p|$, as proposed in this paper. We then have that computing all quantities $\{l_i(x_i), h_i(x_i)\}_{i=1}^{D}$ takes $\mathcal{O}(\log(D) \cdot |p|)$ time. Therefore, en- or decoding $\boldsymbol{x}$ with a streaming code (e.g., Arithmetic Coding) takes $\mathcal{O}(\log(D) \cdot |p| + D) = \mathcal{O}(\log(D) \cdot |p|)$ time.*

The properties of PCs that enable this efficient lossless compression algorithm will be described in Sec. 3.1, and the backbone inference algorithm with $\mathcal{O}(\log(D) \cdot |p|)$ time complexity will later be shown as Alg. 1. Table 1 provides an (incomplete) summary of our empirical results. First, the PC-based lossless compression algorithm is fast and competitive. As shown in Table 1, the small PC model achieved a near-SoTA bitrate while being $\sim 15\times$ faster than other neural compression algorithms with a similar bitrate. Next, PCs can be integrated with Flow-/VAE-based compression methods. As illustrated in Table 1(right), the integrated model significantly improved performance on sub-sampled ImageNet compared to the base IDF model.

## 3 COMPUTATIONALLY EFFICIENT (DE)COMPRESSION WITH PCs

In the previous section, we have boiled down the task of lossless compression to calculating conditional probabilities $\{l_i(x_i), h_i(x_i)\}_{i=1}^{D}$ given $p$ and $x_i$. This section takes PCs into consideration and demonstrates how these queries can be computed efficiently. In the following, we first introduce relevant background on PCs (Sec. 3.1), and then proceed to introduce the PC-based (de)compression algorithm (Sec. 3.2). Finally, we empirically evaluate the optimality and speed of the proposed compressor and decompressor (Sec. 3.3).

---

[2]Flow-model-based neural compression algorithms adopt $p$ defined on mutually independent latent variables (denoted $\mathbf{Z}$), and improve expressiveness by learning bijection functions between $\mathbf{Z}$ and $\mathbf{X}$ (i.e., the input space). This is orthogonal to our approach of directly learn better $p$. Furthermore, we can naturally integrate the proposed expressive $p$ with bijection functions and achieve better performance as demonstrated in Sec. 5.

Table 1: An (incomplete) summary of our empirical results. "Comp." stands for compression.

| Method | MNIST (10,000 test images) | | | Method | ImageNet32 | ImageNet64 |
| | Theoretical bpd | Comp. bpd | En- & decoding time | | Theoretical bpd | Theoretical bpd |
|---|---|---|---|---|---|---|
| PC (small) | 1.26 | 1.30 | **53** | PC+IDF | **3.99** | **3.71** |
| PC (large) | **1.20** | **1.24** | 168 | | | |
| IDF | 1.90 | 1.96 | 880 | IDF | 4.15 | 3.90 |
| BitSwap | 1.27 | 1.31 | 904 | RealNVP | 4.28 | 3.98 |
| | | | | Glow | 4.09 | 3.81 |

## 3.1 BACKGROUND: PROBABILISTIC CIRCUITS

Probabilistic Circuits (PCs) are an umbrella term for a wide variety of Tractable Probabilistic Models (TPMs). They provide a set of succinct definitions for popular TPMs such as Sum-Product Networks (Poon & Domingos, 2011), Arithmetic Circuits (Shen et al., 2016), and Probabilistic Sentential Decision Diagrams (Kisa et al., 2014). The syntax and semantics of a PC are defined as follows.

**Definition 1** (Probabilistic Circuits). A PC $p(\mathbf{X})$ represents a probability distribution over $\mathbf{X}$ via a parametrized directed acyclic graph (DAG) with a single root node $n_r$. Similar to neural networks, every node of the DAG defines a computational unit. Specifically, each leaf node corresponds to an *input unit*; each inner node $n$ represents either a *sum* or a *product unit* that receives inputs from its children, denoted $\mathsf{in}(n)$. Each node $n$ encodes a probability distribution $p_n$, defined as follows:

Figure 1: An example structured-decomposable PC. The feedforward order is from left to right; inputs are assumed to be boolean variables; parameters are labeled on the corresponding edges. Probability of each unit given input assignment $x_1\overline{x_2}x_4$ is labeled blue next to the corresponding unit.

$$p_n(\boldsymbol{x}) := \begin{cases} f_n(\boldsymbol{x}) & \text{if } n \text{ is an input unit,} \\ \sum_{c\in\mathsf{in}(n)} \theta_{n,c} \cdot p_c(\boldsymbol{x}) & \text{if } n \text{ is a sum unit,} \\ \prod_{c\in\mathsf{in}(n)} p_c(\boldsymbol{x}) & \text{if } n \text{ is a product unit,} \end{cases}$$
(2)

where $f_n(\cdot)$ is an univariate input distribution (e.g., Gaussian, Categorical), and $\theta_{n,c}$ denotes the parameter that corresponds to edge $(n,c)$. Intuitively, sum and product units encode weighted mixtures and factorized distributions of their children's distributions, respectively. To ensure that a PC models a valid distribution, we assume the parameters associated with any sum unit $n$ are normalized: $\forall n, \sum_{c\in\mathsf{in}(n)} \theta_{n,c} = 1$. We further assume w.l.o.g. that a PC alternates between sum and product units before reaching an input unit. The size of a PC $p$, denoted $|p|$, is the number of edges in its DAG.

This paper focuses on PCs that can compute arbitrary marginal queries in time linear in their size, since this is necessary to unlock the efficient (de)compression algorithm. In order to support efficient marginalization, PCs need to be *decomposable* (Def. 2),[3] which is a property of the (variable) scope $\phi(n)$ of PC units $n$, that is, the collection of variables defined by all its descendent input units.

**Definition 2** (Decomposability). A PC is decomposable if for every product unit $n$, its children have disjoint scopes: $\forall c_1, c_2 \in \mathsf{in}(n)\,(c_1 \neq c_2), \phi(c_1) \cap \phi(c_2) = \varnothing$.

All product units in Fig. 1 are decomposable. For example, each purple product unit (whose scope is $\{X_1, X_2\}$) has two children with disjoint scopes $\{X_1\}$ and $\{X_2\}$, respectively. In addition to Def. 2, we make use of another property, *structured decomposability*, which is the key to guaranteeing computational efficiency of the proposed (de)compression algorithm.

**Definition 3** (Structured decomposability). A PC is structured-decomposable if (i) it is decomposable and (ii) for every pair of product units $(m,n)$ with identical scope (i.e., $\phi(m) = \phi(n)$), we have that $|\mathsf{in}(m)| = |\mathsf{in}(n)|$ and the scopes of their children are pairwise identical: $\forall i \in \{1, ..., |\mathsf{in}(m)|\}, \phi(cm_i) = \phi(cn_i)$, where $cm_i$ and $cn_i$ are the $i$th child unit of $m$ and $n$.

---

[3]Another property called *smoothness* is also required to compute marginals efficiently. However, since enforcing smoothness on any structured-decomposable PC only imposes at most an almost-linear increase in its size (Shih et al., 2019), we omit introducing it here (all PCs used in this paper are structured-decomposable).

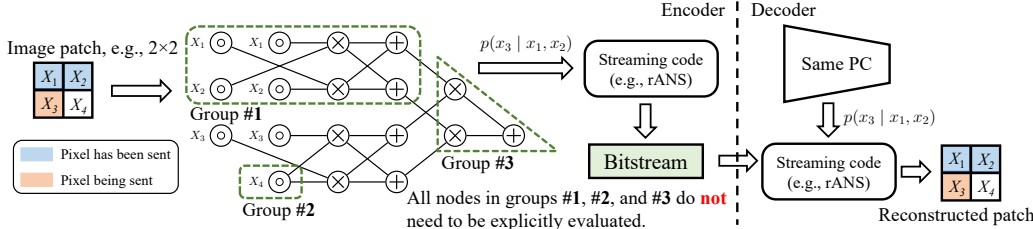

Figure 2: Overview of the PC-based (de)compressor. The encoder's side sequentially compresses variables one-by-one using the conditional probabilities given all sent variables. These probabilities are computed efficiently using Alg. 1. Finally, a streaming code uses conditional probabilities to compress the variables into a bitstream. On the decoder's side, a streaming code decodes the bitstream to reconstruct the image with the conditional probabilities computed by the PC.

The PC shown in Fig. 1 is structured-decomposable because for all three groups of product units with the same scope (grouped by their colors), their children divide the variable scope in the same way. For example, the children of both orange units decompose the scope $\{X_1, X_2, X_3, X_4\}$ into $\{X_1, X_2\}$ and $\{X_3, X_4\}$.

As a key sub-routine in the proposed algorithm, we describe how to compute marginal queries given a smooth and (structured-)decomposable PC in $\mathcal{O}(|p|)$ time. First, we assign probabilities to every input unit: for an input unit $n$ defined on variable $X$, if evidence is provided for $X$ in the query (e.g., $X = x$ or $X < x$), we assign to $n$ the corresponding probability (e.g., $p(X = x)$, $p(X < x)$) according to $f_n$ in Eq. (2); if evidence of $X$ is not given, probability 1 is assigned to $n$. Next, we do a feedforward (children before parents) traverse of inner PC units and compute their probabilities following Eq. (2). The probability assigned to the root unit is the final answer of the marginal query. Concretely, consider computing $p(x_1, \overline{x_2}, x_4)$ for the PC in Fig. 1. This is done by (i) assigning probabilities to the input units w.r.t. the given evidence $x_1$, $\overline{x_2}$, and $x_4$ (assign 0 to the input unit labeled $X_2$ and $\neg X_4$ as they contradict the given evidence; all other input units are assigned probability 1), and (ii) evaluate the probabilities of sum/product units following Eq. (2). Evaluated probabilities are labeled next to the corresponding units, hence the marginal probability at the output is $p(x_1, \overline{x_2}, x_4) = 0.056$.

## 3.2 Efficient (De-)compression With Structured-Decomposable PCs

The proposed PC-based (de)compression algorithm is outlined in Fig. 2. Consider compressing an 2-by-2 image, whose four pixels are denoted as $X_1, \ldots, X_4$. As discussed in Sec. 2, the encoder converts the image into a bitstream by encoding all variables autoregressively. For example, suppose we have encoded $x_1, x_2$. To encode the next variable $x_3$, we compute the left and right side cumulative probability of $x_3$ given $x_1$ and $x_2$, which are defined as $l_3(x_3)$ and $h_3(x_3)$ in Sec. 2, respectively. A streaming code then encodes $x_3$ into a bitstream using these probabilities. Decoding is also performed autoregressively. Specifically, after $x_1$ and $x_2$ are decoded, the same streaming code uses the information from the bitstream and the conditional distribution $p(x_3 \mid x_1, x_2)$ to decode $x_3$.

Therefore, the main computation cost of the above en- and decoding procedures comes from calculating the $2D$ conditional probabilities $\{l_i(x), h_i(x)\}_{i=1}^{D}$ w.r.t. any $\boldsymbol{x}$. Since every conditional probability can be represented as the quotient of two marginals, it is equivalent to compute the two following sets of marginals: $F(\boldsymbol{x}) := \{p(x_1, \ldots, x_i)\}_{i=1}^{D}$ and $G(\boldsymbol{x}) := \{p(x_1, \ldots, x_{i-1}, X_i < x_i)\}_{i=1}^{D}$.

As a direct application of the marginal algorithm described in Sec. 3.1, for every $\boldsymbol{x} \in \mathsf{val}(\mathbf{X})$, computing the $2D$ marginals $\{F(\boldsymbol{x}), G(\boldsymbol{x})\}$ takes $\mathcal{O}(D \cdot |p|)$ time. However, the linear dependency on $D$ would render compression and decompression extremely time-consuming.

We can significantly accelerate the en- and decoding times if the PC is structured-decomposable (see Definition 3). To this end, we introduce an algorithm that computes $F(\boldsymbol{x})$ and $G(\boldsymbol{x})$ in $\mathcal{O}(\log(D) \cdot |p|)$ time (instead of $\mathcal{O}(D \cdot |p|)$), given a smooth and structured-decomposable PC $p$. For ease of presentation, we only discuss how to compute $F(\boldsymbol{x})$ – the values $G(\boldsymbol{x})$ can be computed analogously.[4]

Before proceeding with a formal argument, we give a high-level explanation of the acceleration. In practice, we only need to evaluate a small fraction of PC units to compute each of its $D$ marginals.

---

[4]The only difference between the computation of the $i$th term of $F(\boldsymbol{x})$ and the $i$th term of $G(\boldsymbol{x})$ is in the value assigned to the inputs for variable $X_i$ (i.e., probabilities $p_n(X_i = x)$ vs. $p_n(X_i < x)$).

---

**Algorithm 1** Compute $F(\boldsymbol{x})$ (see Alg. 3 for details)

1: **Input:** A smooth and structured-decomposable PC $p$, variable instantiation $\boldsymbol{x}$
2: **Output:** $F_\pi(\boldsymbol{x}) = \{p(x_1, \ldots, x_i)\}_{i=1}^D$
3: **Initialize:** The probability $p(n)$ of every unit $n$ is initially set to 1
4:        $\forall i, \mathrm{eval}_i \leftarrow$ the set of PC units $n$ that need to be evaluated in the $i$th iteration
5: **for** $i = 1$ **to** $D$ **do**
6: ⌊ Evaluate PC units in $\mathrm{eval}_i$ in a bottom-up manner and compute $p(x_1, \ldots, x_i)$

---

This is different from regular neural networks and the key to speeding up the computation of $F(\boldsymbol{x})$. In contrast to neural networks, changing the input only slightly will leave most activations unchanged for structured-decomposable PCs. We make use of this property by observing that adjacent marginals in $F(\boldsymbol{x})$ only differ in one variable — the $i$th term only adds evidence $x_i$ compared to the $(i-1)$th term. We will show that such similarities between the marginal queries will lead to an algorithm that guarantees $\mathcal{O}(\log(D) \cdot |p|)$ overall time complexity.

An informal version of the proposed algorithm is shown in Alg. 1.[5] In the main loop (lines 5-6), the $D$ terms in $F(\boldsymbol{x})$ are computed one-by-one. Although the $D$ iterations seem to suggest that the algorithm scales linearly with $D$, we highlight that each iteration on average re-evaluates only $\log(D)/D$ of the PC. Therefore, the computation cost of Alg. 1 scales logarithmically w.r.t. $D$. The set of PC units need to be re-evaluated, $\mathrm{eval}_i$, is identified in line 4, and lines 6 evaluates these units in a feedforward manner to compute the target probability (i.e., $p(x_1, \ldots, x_i)$).

Specifically, to minimize computation cost, at iteration $i$, we want to select a set of PC units $\mathrm{eval}_i$ that (i) guarantees the correctness of the target marginal, and (ii) contains the minimum number of units. We achieve this by recognizing three types of PC units that can be safely eliminated for evaluation. Take the PC shown in Fig. 2 as an example. Suppose we want to compute the third term in $F(\boldsymbol{x})$ (i.e., $p(x_1, x_2, x_3)$). First, all PC units in Group #1 do not need to be re-evaluated since their value only depends on $x_1$ and $x_2$ and hence remains unchanged. Next, PC units in Group #2 evaluate to 1. This can be justified from the two following facts: (i) input units correspond to $X_4$ have probability 1 while computing $p(x_1, x_2, x_3)$; (ii) for any sum or product unit, if all its children have probability 1, it also has probability 1 following Eq. (2). Finally, although the activations of the PC units in Group #3 will change when computing $p(x_1, x_2, x_3)$, we do not need to *explicitly* evaluate these units — the root node's probability can be equivalently computed using the weighted mixture of probabilities of units in $\mathrm{eval}_i$. The correctness of this simplification step is justified in Appx. A.1.

The idea of partially evaluating a PC originates from the Partial Propagation (PP) algorithm (Butz et al., 2018). However, PP can only prune away units in Group #2. Thanks to the specific structure of the marginal queries, we are able to also prune away units in Groups #1 and #3.

Finally, we provide additional technical details to rigorously state the complexity of Alg. 1. First, we need the variables $\mathbf{X}$ to have a specific order determined by the PC $p$. To reflect this change, we generalize $F(\boldsymbol{x})$ to $F_\pi(\boldsymbol{x}) := \{p(x_{\pi_1}, \ldots, x_{\pi_i})\}_{i=1}^D$, where $\pi$ defines some variable order over $\mathbf{X}$, i.e., the $i$th variable in the order defined by $\pi$ is $X_{\pi_i}$. Next, we give a technical assumption and then formally justify the *correctness* and *efficiency* of Alg. 1 when using an optimal variable order $\pi^*$.

**Definition 4.** For a smooth structured-decomposable PC $p$ over $D$ variables, for any scope $\phi$, denote $\mathrm{nodes}(p, \phi)$ as the set of PC units in $p$ whose scope is $\phi$. We say $p$ is *balanced* if for every scope $\phi'$ that is equal to the scope of any unit $n$ in $p$, we have $|\mathrm{nodes}(p, \phi')| = \mathcal{O}(|p|/D)$.

**Theorem 1.** *For a smooth structured-decomposable balanced PC $p$ over $D$ variables $\mathbf{X}$ and a sample $\boldsymbol{x}$, there exists a variable order $\pi^*$, s.t. Alg. 3 correctly computes $F_{\pi^*}(\boldsymbol{x})$ in $\mathcal{O}(\log(D) \cdot |p|)$ time.*

*Proof.* First note that Alg. 3 is a detailed version of Alg. 1. The high-level idea of the proof is to first show how to compute the optimal variable order $\pi^*$ for any smooth and structured-decomposable PC. Next, we justify the correctness of Alg. 3 by showing (i) we only need to evaluate units that satisfy the criterion in line 6 of Alg. 3 and (ii) weighing the PC units with the *top-down probabilities* (Appx. A.1) always give the correct result. Finally, we use induction (on $D$) to demonstrate Alg. 3 computes $\mathcal{O}(\log(D) \cdot |p|)$ PC units in total if $\pi^*$ is used. See Appx. A.2 for further details. □

While Def. 4 may seem restrictive at first glance, we highlight that most existing PC structures such as EiNets (Peharz et al., 2020a), RAT-SPNs (Peharz et al., 2020b) and HCLTs (Sec. 4.1) are balanced

---

[5]See Appx. A.1 for the formal algorithm and its detailed elaboration.

Table 2: Efficiency and optimality of the (de)compressor. The compression (resp. decompression) time are the total computation time used to encode (resp. decode) all 10,000 MNIST test samples on a single TITAN RTX GPU. The proposed (de)compressor for structured-decomposable PCs is 5-40x faster than IDF and BitSwap and only leads to a negligible increase in the codeword bpd compared to the theoretical bpd. HCLT is a PC model that will be introduced in Sec. 4.1.

| Method | # parameters | Theoretical bpd | Codeword bpd | Comp. time (s) | Decomp. time (s) |
|---|---|---|---|---|---|
| PC (HCLT, $M=16$) | 3.3M | 1.26 | 1.30 | 9 | 44 |
| PC (HCLT, $M=24$) | 5.1M | 1.22 | 1.26 | 15 | 86 |
| PC (HCLT, $M=32$) | 7.0M | 1.20 | 1.24 | 26 | 142 |
| IDF | 24.1M | 1.90 | 1.96 | 288 | 592 |
| BitSwap | 2.8M | 1.27 | 1.31 | 578 | 326 |

(see Appx. A.3 for justifications). Once all marginal probabilities are calculated, samples $x$ can be en- or decoded autoregressively with any streaming codes in time $\mathcal{O}(\log(D)\cdot|p|)$. Specifically, our implementation adopted the widely used streaming code rANS (Duda, 2013).

## 3.3 EMPIRICAL EVALUATION

We compare the proposed algorithm with competitive Flow-model-based (IDF by Hoogeboom et al. (2019)) and VAE-based (BitSwap by Kingma et al. (2019)) neural compression algorithms using the MNIST dataset. We first evaluate bitrates. As shown in Table 2, the PC (de)compressor achieved compression rates close to its theoretical rate estimate — codeword bpds only have ~0.04 loss w.r.t. the corresponding theoretical bpds. We note that PC and IDF have an additional advantage: their reported bitrates were achieved while compressing one sample at a time; however, BitSwap needs to compress sequences of 100 samples to achieve 1.31 codeword bpd (Kingma et al., 2019).

Next, we focus on efficiency. While achieving a better codeword bpd (i.e., 1.30) compared to IDF and BitSwap, a relatively small PC model (i.e., HCLT, $M=16$) encodes (resp. decodes) images 30x (resp. 10x) faster than both baselines.[6] Furthermore, a bigger PC model ($M=32$) with 7M parameters achieved codeword bpd 1.24, and is still 5x faster than BitSwap and IDF. Note that at the cost of increasing the bitrate, one can significantly improve the en- and decoding efficiency. For example, by using a small VAE model, Townsend et al. (2019) managed to compress and decompress 10,000 binarized MNIST samples in 3.26s and 2.82s, respectively.

**Related work** As hinted by Sec. 2, we seek to directly learn probability distributions $p(\mathbf{X})$ that are expressive and support tractable (de)compression. In contrast, existing Flow-based (van den Berg et al., 2020; Zhang et al., 2021b) and VAE-based (Townsend et al., 2019; Kingma et al., 2019; Ho et al., 2019) neural lossless compression algorithms are based on an orthogonal idea: they adopt simple (oftentimes fully factorized) distributions over a latent space $\mathbf{Z}$ to ensure the tractability of encoding and decoding latent codes $z$, and learn expressive neural networks that "transmit" probability mass from $\mathbf{Z}$ to the feature space $\mathbf{X}$ to compress samples $x$ indirectly. We note that both ideas can be *integrated* naturally: the simple latent distributions used by existing neural compression algorithms can be replaced by expressive PC models. We will further explore this idea in Sec. 5.

## 4 SCALING UP LEARNING AND INFERENCE OF PCS

Being equipped with an efficient (de)compressor, our next goal is to learn PC models that achieve good generative performance on various datasets. Although recent breakthroughs have led to PCs that can generate CelebA and SVHN images (Peharz et al., 2020a), PCs have not been shown to have competitive (normalized) likelihoods on image datasets, which directly influence compression rates. In this section, we show that Hidden Chow-Liu Trees (HCLTs) (Liu & Van den Broeck, 2021), a PC model initially proposed for simple density estimation tasks containing binary features, can be scaled up to achieve state-of-the-art performance on various image datasets. In the following, we first introduce HCLTs and demonstrate how to scale up their learning and inference (for compression) in Sec. 4.1, before providing empirical evidence in Sec. 4.2.

## 4.1 HIDDEN CHOW-LIU TREES

Hidden Chow-Liu Trees (HCLTs) are smooth and structured-decomposable PCs that combine the ability of Chow-Liu Trees (CLTs) (Chow & Liu, 1968) to capture feature correlations and the extra

---

[6]HCLT will be introduced in Sec. 4.1; all algorithms use a CPU implementation of rANS as codec. See Appx. B.4 for more details about the experiments.

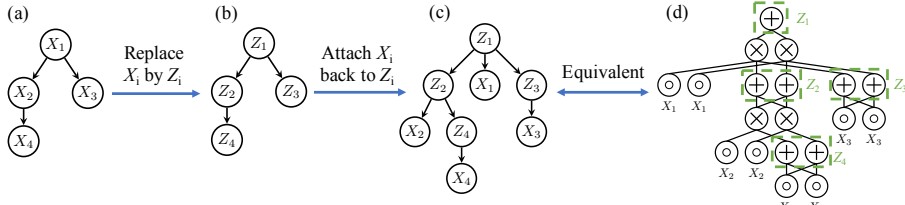

Figure 3: An example of constructing a HCLT PC given a dataset $\mathcal{D}$ with 4 features. (a): Construct the Chow-Liu Tree over variables $X_1, \ldots, X_4$ using $\mathcal{D}$. (b): Replace every variable $X_i$ by its corresponding latent variable $Z_i$. (c): Attach all $X_i$ back to their respective latent variables $Z_i$. (d): This PGM representation of HCLT is compiled into an equivalent PC.

Table 3: Compression performance of PCs on MNIST, FashionMNIST, and EMNIST in bits-per-dimension (bpd). For all neural compression algorithms, numbers in parentheses represent the corresponding theoretical bpd (i.e., models' test-set likelihood in bpd).

| Dataset | HCLT (ours) | IDF | BitSwap | BB-ANS | JPEG2000 | WebP | McBits |
|---|---|---|---|---|---|---|---|
| MNIST | **1.24** (1.20) | 1.96 (1.90) | 1.31 (1.27) | 1.42 (1.39) | 3.37 | 2.09 | (1.98) |
| FashionMNIST | 3.37 (3.34) | 3.50 (3.47) | **3.35** (3.28) | 3.69 (3.66) | 3.93 | 4.62 | (3.72) |
| EMNIST (Letter) | **1.84** (1.80) | 2.02 (1.95) | 1.90 (1.84) | 2.29 (2.26) | 3.62 | 3.31 | (3.12) |
| EMNIST (ByClass) | **1.89** (1.85) | 2.04 (1.98) | 1.91 (1.87) | 2.24 (2.23) | 3.61 | 3.34 | (3.14) |

expressive power provided by latent variable models. Every HCLT can be equivalently represented as a Probabilistic Graphical Model (PGM) (Koller & Friedman, 2009) with latent variables. Specifically, Fig. 3(a)-(c) demonstrate how to construct the PGM representation of an example HCLT. Given a dataset $\mathcal{D}$ containing 4 features $\mathbf{X} = X_1, \ldots, X_4$, we first learn a CLT w.r.t. $\mathbf{X}$ (Fig. 3(a)). To improve expressiveness, latent variables are added to the CLT by the two following steps: (i) replace observed variables $X_i$ by their corresponding latent variables $Z_i$, which are defined to be categorical variables with $M$ (a hyperparameter) categories (Fig. 3(b)); (ii) connect observed variables $X_i$ with the corresponding latent variables $Z_i$ by directed edges $Z_i \rightarrow X_i$. This leads to the PGM representation of the HCLT shown in Fig. 3(c). Finally, we are left with generating a PC that represents an equivalent distribution w.r.t. the PGM in Fig. 3(c), which is detailed in Appx. B.2. Fig. 3(d) illustrates an HCLT that is equivalent to the PGM shown in Fig. 3(c) (with $M = 2$).

Recent advances in scaling up learning and inference of PCs largely rely on the regularity of the PC architectures they used (Peharz et al., 2020a;b) — the layout of the PCs can be easily *vectorized*, allowing them to use well-developed deep learning packages such as PyTorch (Paszke et al., 2019). However, due to the irregular structure of learned CLTs, HCLTs cannot be easily vectorized. To overcome this problem, we implemented customized GPU kernels for parameter learning and marginal query computation (i.e., Alg. 3) based on Juice.jl (Dang et al., 2021), an open-source Julia package. The kernels automatically segment PC units into layers such that the computation in every layer can be fully parallelized. As a result, we can train PCs with millions of parameters in less than an hour and en- or decode samples very efficiently. Implementation details can be found in Appx. B.3.

**Related work** Finding good PC architectures has been a central topic in the literature (Choi et al., 2020). A recent trend for learning smooth and (structured-)decomposable PCs is to construct large models with pre-defined architecture, which is mainly determined by the variable ordering strategies. For example, RAT-SPNs (Peharz et al., 2020b) and EiNets (Peharz et al., 2020a) use random variable orders, Gens & Domingos (2013) proposes an effective variable ordering for image data, and other works propose data-dependent orderings based on certain information criterion (Rooshenas & Lowd, 2014) or clustering algorithms (Gens & Domingos, 2013). Alternatively, researchers have focused on methods that iteratively grow PC structures to better fit the data (Dang et al., 2020; Liang et al., 2017).

## 4.2 EMPIRICAL EVALUATION

Bringing together expressive PCs (i.e., HCLTs) and our (de)compressor, we proceed to evaluate the compression performance of the proposed PC-based algorithm. We compare with 5 competitive lossless compression algorithm: JPEG2000 (Christopoulos et al., 2000); WebP; IDF (Hoogeboom et al., 2019), a Flow-based lossless compression algorithm; BitSwap (Kingma et al., 2019), BB-ANS (Townsend et al., 2018), and McBits (Ruan et al., 2021), three VAE-based lossless compression methods. All 6 methods were tested on 4 datasets, which include MNIST (Deng, 2012), FashionMNIST

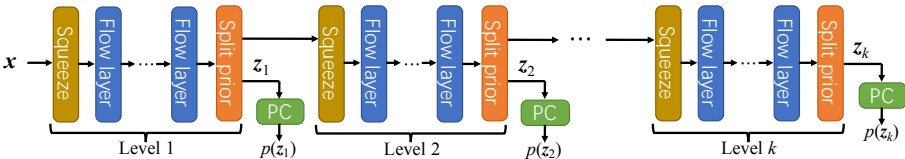

Figure 4: Using PCs as prior distributions of the IDF model (Hoogeboom et al., 2019). PCs are used to represent the $k$ sets of latent variables $\{z_i\}_{i=1}^{k}$.

(Xiao et al., 2017), and two splits of EMNIST (Cohen et al., 2017). As shown in Table 3, the proposed method out-performed all 5 baselines in 3 out of 4 datasets. On FashionMNIST, where the proposed approach did not achieve a state-of-the-art result, it was only $0.02$ bpd worse than BitSwap.

## 5 PCs AS EXPRESSIVE PRIOR DISTRIBUTIONS OF FLOW MODELS

As hinted by previous sections, PCs can be naturally integrated with existing neural compression algorithms: the simple latent variable distributions used by Flow- and VAE-based lossless compression methods can be replaced by more expressive distributions represented by PCs. In this section, we take IDF (Hoogeboom et al., 2019), a Flow-based lossless compression model, as an example to demonstrate the effectiveness of such model

Table 4: Theoretical bpd of 5 Flow-based generative models on three natural image datasets.

| Model | CIFAR10 | ImageNet32 | ImageNet64 |
|---|---|---|---|
| RealNVP | 3.49 | 4.28 | 3.98 |
| Glow | 3.35 | 4.09 | 3.81 |
| IDF | 3.32 | 4.15 | 3.90 |
| IDF++ | **3.24** | 4.10 | 3.81 |
| PC+IDF | 3.28 | **3.99** | **3.71** |

integration. IDF was chosen because its authors provided an open-source implementation on GitHub. In theory, PC can be integrated with any VAE- and Flow-based model.

The integrated model is illustrated in Fig. 4. Following Hoogeboom et al. (2019), an IDF model contains $k$ levels. Each level contains a squeeze layer (Dinh et al., 2016), followed by several integer flow layers and a prior layer. Each level $i$ outputs a set of latent variables $z_i$, which are originally defined as a set of mutually independent discretized logistic variables (Kingma et al., 2016). Instead, we propose to model every set of latent variables $z_i$ with a PC $p(z_i)$. Specifically, we adopted the EiNet codebase (Peharz et al., 2020a) and used a PC structure similar to the one proposed by Gens & Domingos (2013). We adopted the discretized logistic distribution for all leaf units in the PCs. Given a sample $x$, the log-likelihood of the model is the sum of the $k$ PCs' output log-likelihood: $\log p(x) = \sum_{i=1}^{k} \log p(z_i \mid x)$. Since both IDF and the PC models are fully differentiable, the PC+IDF model can be trained end-to-end via gradient descent. Details regarding model architecture and parameter learning are provided in Appx. B.5.

We proceed to evaluate the generative performance of the proposed PC+IDF model on 3 natural image datasets: CIFAR10, ImageNet32, and ImageNet64. Results are shown in Table 4. First, compared to 4 baselines (i.e., IDF, IDF++ (van den Berg et al., 2020), Glow (Kingma & Dhariwal, 2018), and RealNVP (Dinh et al., 2016)), PC+IDF achieved the best bpd on ImageNet32 and ImageNet64. Next, PC+IDF improved over its base model IDF by 0.04, 0.16, and 0.19 bpd on three datasets, respectively. This shows the benefit of integrating PCs with IDFs. Although not tested in our experiments, we conjecture that the performance could be further improved by integrating PCs with better Flow models (e.g., IDF++). Concurrently, Zhang et al. (2021a) proposes an autoregressive model-based compressor NeLLoC, which achieved SoTA results on natural image datasets including CIFAR-10.

Compression and decompression with the PC+IDF model can be done easily: we can adopt the high-level compression algorithm of IDF and replace the parts of en- or decoding latent variables $z_i$ with the proposed PC (de)compressor. Improving the compression performance of these hybrid models is left for future work. Note that Thm. 1 only applies to the PC component, and the compression time still depends linearly on the size of the neural network.

## 6 CONCLUSIONS

This paper proposes to use Probabilistic Circuits (PCs) for lossless compression. We develop a theoretically-grounded (de)compression algorithm that efficiently encodes and decodes close to the model's theoretical rate estimate. Our work provides evidence that more "niche" generative model architectures such as PCs can make valuable contributions to neural compression.

**Acknowledgements** Guy Van den Broeck acknowledges funding by NSF grants #IIS-1943641, #IIS-1956441, #CCF-1837129, DARPA grant #N66001-17-2-4032, and a Sloan Fellowship. Stephan Mandt acknowledges funding by NSF grants #IIS-2047418 and #IIS-2007719. We thank Yibo Yang for feedback on the manuscript's final version.

**Ethics and Reproducibility Statement** We are not aware of any ethical concerns of our research. To facilitate reproducibility, we have uploaded our code to the following GitHub repo: `https://github.com/Juice-jl/PressedJuice.jl`. In addition, we have provided detailed algorithm tables Alg. 2 and 3 for all proposed algorithms, and elaborated each step in detail in the main text (Sec. 3). Formal proofs of all theorems, and details of all experiments (e.g., hardware specifications, hyperparameters) are provided in the appendix.

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

# Supplementary Material

## A  ALGORITHM DETAILS AND PROOFS

This section provides additional details about the algorithm used to compute the conditional probabilities $F_\pi(\boldsymbol{x})$ (i.e., Alg. 1) and the full proof of the theorems stated in the main paper.

### A.1  DETAILS OF ALG. 1

This section provides additional technical details of Alg. 1. Specifically, we demonstrate (i) how to select the set of PC units $\mathrm{eval}_i$ (cf. Alg. 1 line 5) and (ii) how to compute $p(x_1, \dots, x_i)$ as a weighted mixture of $P_i$ (cf. Alg. 1 line 7). Using the example in Fig. 5, we aim to provide an intuitive illustration to both problems. As an extension to Alg. 1, rigorous and executable pseudocode for the proposed algorithm can be found in Alg. 2 and 3.

The key to speeding up the naive marginalization algorithm is the observation that we only need to evaluate a small fraction of PC units to compute each of the $D$ marginals in $F_\pi(\boldsymbol{x})$. Suppose we want to compute $F_\pi(\boldsymbol{x})$ given the structured-decomposable PC shown in Fig. 5(a), where $\oplus$, $\otimes$, and $\odot$ denote sum, product, and input units, respectively. Model parameters are omitted for simplicity. Consider using the variable order $\pi = (X_1, X_2, X_3)$ (Fig. 5(b)). We ask the following question: what is the minimum set of PC units that need to be evaluated in order to compute $p(X_1 = x_1)$ (the first term in $F_\pi(\boldsymbol{x})$)? First, every PC unit with scope $\{X_1\}$ (i.e., the two nodes colored blue) has to be evaluated. Next, every PC unit $n$ that is not an ancestor of the two blue units (i.e., "non-ancestor units" in Fig. 5(b)) must have probability 1 since (i) leaf units correspond to $X_2$ and $X_3$ have probability 1 while computing $p(X_1 = x_1)$, and (ii) for any sum or product unit, if all its children have probability 1, it also has probability 1 following Eq. (2). Therefore, we do not need to evaluate these non-ancestor units. Another way to identify these non-ancestor units is by inspecting their variable scopes — if the variable scope of a PC unit $n$ does not contain $X_1$, it must has probability 1 while computing $p(X_1 = x_1)$. Finally, following all ancestors of the two blue units (i.e., "ancestor units" in Fig. 5(b)), we can compute the probability of the root unit, which is the target quantity $p(X_1 = x_1)$. At a first glance, this seems to suggest that we need to evaluate these ancestor units explicitly. Fortunately, as we will proceed to show, the root unit's probability can be equivalently computed using the blue units' probabilities weighted by a set of cached *top-down probabilities*.

For ease of presentation, denote the two blue input units as $n_1$ and $n_2$, respectively. A key observation is that the probability of every ancestor unit of $\{n_1, n_2\}$ (including the root unit) can be represented as a weighted mixture over $p_{n_1}(\boldsymbol{x})$ and $p_{n_2}(\boldsymbol{x})$, the probabilities assigned to $n_1$ and $n_2$, respectively. The reason is that for each decomposable product node $m$, only distributions defined on disjoint variables shall be multiplied. Since $n_1$ and $n_2$ have the same variable scope, their distributions will not be multiplied by any product node. Following the above intuition, the top-down probability $p_{\mathrm{down}}(n)$ of PC unit $n$ is designed to represent the "weight" of $n$ w.r.t. the probability of the root unit. Formally, $p_{\mathrm{down}}(n)$ is defined as the sum of the probabilities of every path from $n$ to the root unit $n_r$, where the probability of a path is the product of all edge parameters traversed by it. Back to our example, using the top-down probabilities, we can compute $p(X_1 = x_1) = \sum_{i=1}^{2} p_{\mathrm{down}}(n_i) \cdot p_{n_i}(x_1)$ without explicitly evaluating the ancestors of $n_1$ and $n_2$. The quantity $p_{\mathrm{down}}(n)$ of all PC units $n$ can be computed by Alg. 2 in $\mathcal{O}(|p|)$ time. Specifically, the algorithm performs a top-down traversal over all PC units $n$, and updates the top-down probabilities of their children $\mathrm{in}(n)$ along the process.

Therefore, we only need to compute the two PC units with scope $\{X_1\}$ in order to calculate $p(X_1 = x_1)$. Next, when computing the second term $p(X_1 = x_1, X_2 = x_2)$, as illustrated in Fig. 5(b), we can reuse the evaluated probabilities of $n_1$ and $n_2$, and similarly only need to evaluate the PC units with scope $\{X_2\}$, $\{X_2, X_3\}$, or $\{X_1, X_2, X_3\}$ (i.e., nodes colored purple). The same scheme can be used when computing the third term, and we only evaluate PC units with scope $\{X_3\}$, $\{X_2, X_3\}$, or $\{X_1, X_2, X_3\}$ (i.e., all red nodes). As a result, we only evaluate 20 PC units in total, compared to $3 \cdot |p| = 39$ units required by the naive approach.

This procedure is formalized in Alg. 3, which adds additional technical details compared to Alg. 1. In the main loop (lines 5-9), the $D$ terms in $F_\pi(\boldsymbol{x})$ are computed one-by-one. While computing each term, we first find the PC units that need to be evaluated (line 6).[7] After computing their probabilities

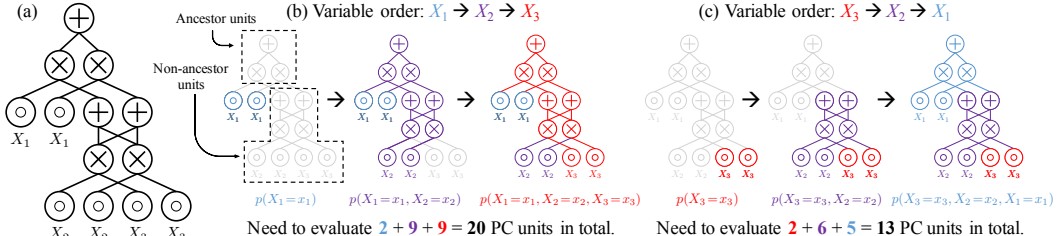

Figure 5: Good variable orders lead to more efficient computation of $F_\pi(\boldsymbol{x})$. Consider the PC $p$ shown in (a). (b): If variable order $X_1, X_2, X_3$ is used, we need to evaluate 20 PC units in total. (c): The optimal variable order $X_3, X_2, X_1$ allows us to compute $F_\pi(\boldsymbol{x})$ by only evaluating 13 PC units.

---

**Algorithm 2** PC Top-down Probabilities

---

1: **Input:** A smooth and structured-decomposable PC $p$
2: **Output:** The top-down probabilities $p_{\text{down}}(n)$ of all PC units $n$
3: For every PC unit $n$ in $p$, initialize $p_{\text{down}}(n) \leftarrow 0$
4: **foreach** unit $n$ traversed in preorder (parent before children) **do**
5:     **if** $n$ is the root node of $p$ **then** $p_{\text{down}}(n) \leftarrow 1$
6:     **elif** $n$ is a sum unit **then foreach** $c \in \text{in}(n)$ **do** $p_{\text{down}}(c) \leftarrow p_{\text{down}}(c) + p_{\text{down}}(n) \cdot \theta_{n,c}$
7:     **elif** $n$ is a product unit **then foreach** $c \in \text{in}(n)$ **do** $p_{\text{down}}(c) \leftarrow p_{\text{down}}(c) + p_{\text{down}}(n)$

---

in a bottom-up manner (line 7), we additionally use the pre-computed top-down probabilities to obtain the target marginal probability (lines 8-9).

The previous example demonstrates that even without a careful choice of variable order, we can significantly lower the computation cost by only evaluating the necessary PC units. We now show that with an *optimal* choice of variable order (denoted $\pi^*$), the cost can be further reduced. Consider using order $\pi^* = (X_3, X_2, X_1)$, as shown in Fig. 5(c), we only need to evaluate $2+6+5=13$ PC units in total when running Alg. 3. This optimal variable order is the key to guaranteeing $\mathcal{O}(\log(D)\cdot|p|)$ computation time. In the following, we first give a technical assumption and then proceed to justify the *correctness* and *efficiency* of Alg. 3 when using the optimal variable order $\pi^*$.

## A.2   Proof of Theorem 1

As hinted by the proof sketch given in the main text, this proof consists of three main parts — (i) construction of the optimal variable order $\pi^*$ given a smooth and structured-decomposable PC, (ii) justify the correctness of Alg. 3, and (iii) prove that $F_{\pi^*}(\boldsymbol{x})$ can be computed by evaluating no more than $\mathcal{O}(\log(K)\cdot|p|)$ PC units (i.e., analyze the time complexity of Alg. 3).

**Construction of an optimal variable order**   For ease of illustration, we first transform the original smooth and structured-decomposable PC into an equivalent PC where every product node has two children. Fig. 6 illustrates this transformation on any product node with more than two children. Note that this operation will not change the number of parameters in a PC, and will only incur at most $2\cdot|p|$ edges.

We are now ready to define the variable tree (*vtree*) (Kisa et al., 2014) of a smooth and structured-decomposable PC. Specifically, a *vtree* is a binary tree structure whose leaf nodes are labeled with a PC's input features/variables $\mathbf{X}$ (every leaf node is labeled with one variable). A PC conforms to a vtree if for every product unit $n$, there is a corresponding vtree node $v$ such that children of $n$ split the variable scope $\phi(n)$ in the same way as the children of the vtree node $v$. According to its definition, every smooth and structured-decomposable PC whose product units all have two children must conform to a vtree (Kisa et al., 2014). For example, the PC shown in Fig. 7(a) conforms to the vtree illustrated in Fig. 7(b). Similar to PCs, we define the scope $\phi(v)$ of a vtree node $v$ as the set of all descendent leaf variables of $v$.

We say that a unit $n$ in a smooth and structured-decomposable PC conforms to a node $v$ in the PC's corresponding vtree if their scopes are identical. For ease of presentation, define $\varphi(p, v)$ as the set of PC units that conform to vtree node $v$. Additionally, we define $\varphi_{\text{sum}}(p, v)$ and $\varphi_{\text{prod}}(p, v)$ as the set of sum and product units in $\varphi(p, v)$, respectively.

---

**Algorithm 3** Compute $F_\pi(\boldsymbol{x})$

---

1: **Input:** A smooth and structured-decomposable PC $p$, variable order $\pi$, variable instantiation $\boldsymbol{x}$
2: **Output:** $F_\pi(\boldsymbol{x}) = \{p(x_{\pi_1}, \ldots, x_{\pi_i})\}_{i=1}^D$
3: **Initialize:** The probability $p(n)$ of every unit $n$ is initially set to 1
4: $p_{\text{down}} \leftarrow$ the top-down probability of every PC unit $n$ (i.e., Alg. 2)
5: **for** $i = 1$ **to** $D$ **do**   # Compute the $i$th term in $F_\pi(\boldsymbol{x})$: $p(x_{\pi_1}, \ldots, x_{\pi_i})$
6:    $\text{eval}_i \leftarrow$ the set of PC units $n$ with scopes $\phi(n)$ that satisfy at least one of the following conditions:
      (i) $\phi(n) = \{X_{\pi_i}\}$;    (ii) $n$ is a sum unit and at least one child $c$ of $n$ needs evaluation, i.e., $c \in \text{eval}_i$;
      (iii) $n$ is a product unit and $X_{\pi_i} \in \phi(n)$ and $\nexists c \in \text{in}(n)$ such that $\{X_{\pi_j}\}_{j=1}^i \in \phi(c)$
7:    Evaluate PC units in $\text{eval}_i$ in a bottom-up manner to compute $\{p_n(\boldsymbol{x}) : n \in \text{eval}_i\}$
8:    $\text{head}_i \leftarrow$ the set of PC units in $\text{eval}_i$ such that none of their parents are in $\text{eval}_i$
9:    $p(x_{\pi_1}, \ldots, x_{\pi_i}) \leftarrow \sum_{n \in \text{head}_i} p_{\text{down}}(n) \cdot p_n(\boldsymbol{x})$

---

Next, we define an operation that changes a vtree into an *ordered vtree*, where for each inner node $v$, its left child has more descendent leaf nodes than its right child. See Fig. 7(c-d) as an example. The vtree in Fig. 7(b) is transformed into an ordered vtree illustrated in Fig. 7(c); the corresponding PC (Fig. 7(a)) is converted into an *ordered PC* (Fig. 7(d)). This transformation can be performed by all smooth and structured-decomposable PCs.

We are ready to define the optimal variable order. For a pair of ordered PC and ordered vtree, the optimal variable order $\pi^*$ is defined as the order the leaf vtree nodes (each corresponds to a variable) are accessed following an inorder traverse of the vtree (left child accessed before right child).

**Correctness of Algorithm 3**    Assume we have access to a smooth, structured-decomposable, and ordered PC $p$ and its corresponding vtree. Recall from the above construction, the optimal variable order $\pi^*$ is the order following an inorder traverse of the vtree.

We show that it is sufficient to only evaluate the set of PC units stated in line 6 of Alg. 3. Using our new definition of vtrees, we state line 6 in the following equivalent way. At iteration $i$ (i.e., we want to compute the $i$th term in $F_\pi(\boldsymbol{x})$: $p(x_{\pi_1}, \ldots, x_{\pi_i})$), we need to evaluate all PC units that conform to any vtree node in the set $T_{p,i}$. Here $T_{p,i}$ is defined as the set of vtree nodes $v$ that satisfy the following condition: $X_{\pi_i} \in \phi(v)$ and there does not exist a child $c$ of $v$ such that $\{X_{\pi_j}\}_{j=1}^i \in \phi(c)$. For ease of presentation, we refer to evaluate PC units $\varphi(p, v)$ when we say "evaluate a vtree node $v$".

First, we don't need to evaluate vtree units $v$ where $X_{\pi_i} \notin \phi(v)$ because the probability of these PC units will be identical to that at iteration $i - 1$ (i.e., when computing $p(x_{\pi_1}, \ldots, x_{\pi_{i-1}})$). Therefore, we only need to cache these probabilities computed in previous iterations.

Second, we don't need to evaluate vtree units $v$ where at least one of its children $c$ satisfy $\{X_{\pi_j}\}_{j=1}^{i-1} \in \phi(c)$ because we can obtain the target marginal probability $p(x_{\pi_1}, \ldots, x_{\pi_i})$ following lines 7-9 of Alg. 3. We proceed to show how this is done in the following.

Denote the "highest" in $T_{p,i}$ as $v_{r,i}$ (i.e., the parent of $v_{r,i}$ is not in $T_{p,i}$). According to the variable order $\pi^*$, $v_{r,i}$ uniquely exist for any $i \in [D]$. According to Alg. 2, the top-down probabilities of PC units is defined as follows

- $p_{\text{down}}(n_r) = 1$, where $n_r$ is the PC's root unit.

- For any product unit $n$, $p_{\text{down}}(n) = \sum_{m \in \text{par}(n)} p_{\text{down}}(m) \cdot \theta_{m,n}$, where $\text{par}(n)$ is the set of parent (sum) units of $n$.

- For any sum unit $n$, $p_{\text{down}}(n) = \sum_{m \in \text{par}(n)} p_{\text{down}}(m)$, where $\text{par}(n)$ is the set of parent (product) units of $n$.

We now prove that

$$p(x_{\pi_1}, \ldots, x_{\pi_i}) = \sum_{n \in \varphi_{\text{sum}}(p,v)} p_{\text{down}}(n) \cdot p_n(\boldsymbol{x}) \tag{3}$$

holds when $v = v_{r,i}$.

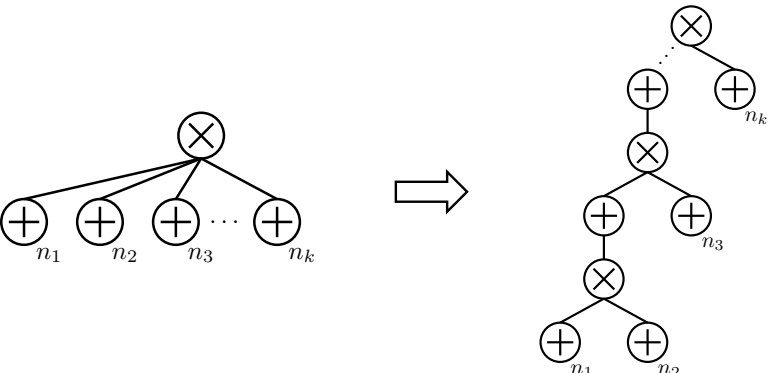

Figure 6: Convert a product unit with $k$ children into an equivalent PC where every product node has two children.

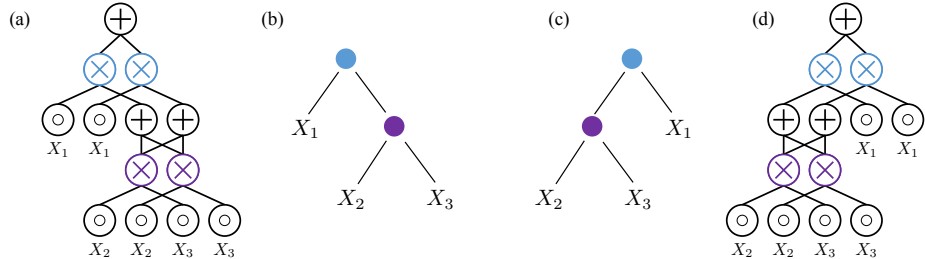

Figure 7: (a-b): An example structured-decomposable PC and a corresponding *vtree*. (c-d): Converting (b) into an ordered vtree. (d) The converted ordered PC that is equivalent to (a).

• **Base case:** If $v$ is the vtree node correspond to $n_r$, then $\varphi_{\text{sum}}(p,v) = \{n_r\}$ and it is easy to verify that

$$p(x_{\pi_1}, \ldots, x_{\pi_i}) = p_{\text{down}}(n_r) \cdot p_{n_r}(\boldsymbol{x}) = \sum_{n \in \varphi_{\text{sum}}(p,v)} p_{\text{down}}(n) \cdot p_n(\boldsymbol{x})$$

• **Inductive case:** Suppose $v$ is an ancestor of $v_{r,i}$ and the parent vtree node $v_p$ of $v$ satisfy Eq. (3). We have

$$p(x_{\pi_1}, \ldots, x_{\pi_i}) = \sum_{m \in \varphi_{\text{sum}}(p,v_p)} p_{\text{down}}(m) \cdot p_m(\boldsymbol{x})$$

$$= \sum_{m \in \varphi_{\text{sum}}(p,v_p)} \sum_{n \in \text{in}(m)} p_{\text{down}}(m) \cdot \theta_{m,n} \cdot p_n(\boldsymbol{x})$$

$$\overset{(a)}{=} \sum_{n \in \varphi_{\text{prod}}(p,v_p)} \underbrace{\sum_{m \in \text{par}(n)} p_{\text{down}}(m) \cdot \theta_{m,n}} _{p_{\text{down}}(n)} \cdot p_n(\boldsymbol{x})$$

$$= \sum_{n \in \varphi_{\text{prod}}(p,v_p)} p_{\text{down}}(n) \cdot p_n(\boldsymbol{x})$$

$$\overset{(b)}{=} \sum_{n \in \varphi_{\text{prod}}(p,v_p)} \sum_{o \in \{o: o \in \text{in}(n), \{X_j\}_{j=1}^i \in \phi(o)\}} p_{\text{down}}(n) \cdot p_o(\boldsymbol{x})$$

$$\overset{(c)}{=} \sum_{o \in \varphi_{\text{sum}}(p,v)} \underbrace{\sum_{n \in \text{par}(o)} p_{\text{down}}(n) \cdot p_o(\boldsymbol{x})}_{p_{\text{down}}(o)}$$

$$= \sum_{o \in \varphi_{\text{sum}}(p,v)} p_{\text{down}}(o) \cdot p_o(\boldsymbol{x})$$

where $(a)$ reorders the terms for summation; $(b)$ holds since $\forall n \in \varphi_{\text{prod}}(p, v_p)$, $p_n(\boldsymbol{x}) = \prod_{o \in \text{in}(n)} p_o(\boldsymbol{x})$ and $\forall o \in \text{in}(n)$ such that $\{X_j\}_{j=1}^i \cap \phi(o) = \varnothing$, $p_o(\boldsymbol{x}) = 1$;[8] $(c)$ holds because

$$\bigcup_{n \in \varphi_{\text{prod}}(p,v_p)} \{o : o \in \text{in}(n), \{X_j\}_{j=1}^i \in \phi(o)\} = \varphi_{\text{sum}}(p, v).$$

Thus, we have prove that Eq. (3) holds for $v = v_{r,i}$, and hence the probability $p(x_{\pi_1}, \ldots, x_{\pi_i})$ can be computed by weighting the probability of PC units $\varphi_{\text{sum}}(p, v_{r,i})$ (line 8 in Alg. 3) with the corresponding top-down probabilities (line 9 in Alg. 3).

**Efficiency of following the optimal variable order**    We proceed to show that when using the optimal variable order $\pi^*$, Alg. 3 evaluates no more than $\mathcal{O}(\log(D) \cdot |p|)$ PC units.

According to the previous paragraphs, whenever Alg. 3 evaluates a PC unit $n$ w.r.t. vtree node $v$, it will evaluate all PC units in $\varphi(p, v)$. Therefore, we instead count the total number of vtree nodes need to be evaluated by Alg. 3. Since the PC is assumed to be balanced Def. 4, for every $v$, we have $\varphi(p, v) = \mathcal{O}(|p|/D)$. Therefore, we only need to show that Alg. 3 evaluates $\mathcal{O}(D \cdot \log(D))$ vtree nodes in total.

We start with the base case, which is PCs correspond to a single vtree leaf node $v$. In this case, $F_{\pi^*}(\boldsymbol{x})$ boils down to computing a single marginal probability $p(x_{\pi_1^*})$, which needs to evaluate PC units $\varphi(p, v)$ once.

Define $f(x)$ as the number of vtree nodes need to be evaluated given a PC corresponds to a vtree node with $x$ descendent leaf nodes. From the base case we know that $f(1) = 1$.

Next, consider the inductive case where $v$ is an inner node that has $x$ descendent leaf nodes. Define the left and right child node of $v$ as $c_1$ and $c_2$, respectively. Let $c_1$ and $c_2$ have $y$ and $z$ descendent leaf nodes, respectively. We want to compute $F_{\pi^*}(\boldsymbol{x})$, which can be broken down into computing two following sets of marginals:

$$\text{Set 1: } \{p(x_{\pi_1^*}, \cdots, x_{\pi_i^*})\}_{i=1}^y, \qquad \text{Set 2: } \{p(x_{\pi_1^*}, \cdots, x_{\pi_i^*})\}_{i=y+1}^{y+z}.$$

Since $\pi^*$ follows the in-order traverse of $v$, to compute the first term, we only need to evaluate $c_1$ and its descendents, that is, we need to evaluate $f(y)$ vtree nodes. This is because the marginal probabilities in set 1 are only defined on variables in $\phi(c_1)$. To compute the second term, in addition to evaluating PC units corresponding to $c_2$ (that is $f(z)$ vtree nodes in total),[9] we also need to re-evaluate the PC units $\varphi(p, v)$ every time, which means we need to evaluate $z$ more vtree nodes. In summary, we need to evaluate

$$f(x) = f(y) + f(z) + z \quad (y \geq z, y + z = x)$$

vtree nodes.

To complete the proof, we upper bound the number of vtree nodes need to be evaluated. Define $g(\cdot)$ as follows:

$$g(x) = \max_{y \in \{1, \ldots, \lfloor \frac{x}{2} \rfloor\}} y + g(y) + g(x - y).$$

It is not hard to verify that $\forall x \in \mathbb{Z}, g(x) \geq f(x)$. Next, we prove that

$$\forall x \in \mathbb{Z} \, (x \geq 2), \, g(x) \leq 3x \log x.$$

First, we can directly verify that $g(2) \leq 3 \cdot 2 \log_2 2 \approx 4.1$. Next, for $x \geq 3$,

$$g(x) = \max_{y \in \{1, \ldots, \lfloor \frac{x}{2} \rfloor\}} y + g(y) + g(x - y)$$

---

[8]This is because the scope of these PC units does not contain any of the variables in $\{X_{\pi_j}\}_{j=1}^i$.

[9]As justified in the second part of this proof, all probabilities of PC units that conform to descendents of $c_1$ will be unchanged when computing the marginals in set 2. Hence we only need to cache these probabilities.

$$\leq \max_{y \in \{1,\dots,\lfloor \frac{x}{2} \rfloor\}} \underbrace{y + 3y \log y + 3(x-y) \log(x-y)}_{h(y)}$$

$$\stackrel{(a)}{\leq} \max\left(1 + 3(x-1)\log(x-1), \left\lfloor \frac{x}{2} \right\rfloor + 3 \left\lfloor \frac{x}{2} \right\rfloor \log \left\lfloor \frac{x}{2} \right\rfloor + 3\left(x - \left\lfloor \frac{x}{2} \right\rfloor\right) \log \left(x - \left\lfloor \frac{x}{2} \right\rfloor\right)\right)$$

$$\leq \max\left(1 + 3(x-1)\log(x-1), \left\lfloor \frac{x}{2} \right\rfloor + 3(x+1)\log \frac{x+1}{2}\right)$$

$$\leq 3x \log x,$$

where $(a)$ holds since according to its derivative, $h(y)$ obtains its maximum value at either $y = 1$ or $y = \left\lfloor \frac{x}{2} \right\rfloor$.

For a structured-decomposable PC with $D$ variables, $g(D) \leq 3D \log D$ vtree nodes need to be evaluated. Since each vtree node corresponds to $\mathcal{O}(\frac{|p|}{D})$ PC units, we need to evaluate $\mathcal{O}(\log(D) \cdot |p|)$ PC units to compute $F_{\pi^*}(\boldsymbol{x})$.

### A.3 HCLTs, EiNets, and RAT-SPNs are Balanced

Consider the compilation from a PGM to an HCLT (Sec. 4.1). We first note that each PGM node $g$ uniquely corresponds to a variable scope $\phi$ of the PC. That is, all PC units correspond to $g$ have the same variable scope. Please first refer to Appx. B.2 for details on how to generate a HCLT given its PGM representation.

In the main loop of Alg. 4 (lines 5-10), for each PGM node $g$ such that $\mathrm{var}(g) \in \mathbf{Z}$, the number of computed PC units are the same ($M$ product units compiled in line 9 and $M$ sum units compiled in line 10). Therefore, for any variable scopes $\phi_1$ and $\phi_2$ possessed by some PC units, we have $|\mathrm{nodes}(p, \phi(m))| \approx |\mathrm{nodes}(p, \phi(n))|$. Since there are in total $\Theta(D)$ different variable scopes in $p$, we have: for any scope $\phi'$ exists in an HCLT $p$, $\mathrm{nodes}(p, \phi') = \mathcal{O}(|p|/D)$.

EiNets and RAT-SPNs are also balanced since they also have an equivalent PGM representation of their PCs. The main difference between these models and HCLTs is the different variable splitting strategy in the product units.

## B Methods and Experiment Details

### B.1 Learning HCLTs

**Computing Mutual Information** As mentioned in the main text, computing the pairwise mutual information between variables $\mathbf{X}$ is the first step to compute the Chow-Liu Tree. Since we are dealing with categorical data (e.g., 0-255 for pixels), we compute mutual information by following its definition:

$$I(X; Y) = \sum_{i=1}^{C_X} \sum_{j=1}^{C_Y} P(X = i, Y = j) \log_2 \frac{P(X = i, Y = j)}{P(X = i)P(Y = j)},$$

where $C_X$ and $C_Y$ are the number of categories for variables $X$ and $Y$, respectively. To lower the computation cost, for image data, we truncate the data by only using 3 most-significant bits. That is, we treat the variables as categorical variables with $2^3 = 8$ categories during the construction of the CLT. Note that we use the full data when constructing/learning the PC.

**Training pipeline** We adopt two types of EM updates — mini-batch and full-batch. In mini-batch EM, parameters are updated according to a step size $\eta$: $\boldsymbol{\theta}^{(k+1)} \leftarrow (1-\eta)\boldsymbol{\theta}^{(k)} + \eta \boldsymbol{\theta}^{(\mathrm{new})}$, where $\boldsymbol{\theta}^{(\mathrm{new})}$ is the EM target computed with a batch of samples; full-batch EM updates the parameters by the EM target computed using the whole dataset. In this paper, HCLTs are trained by first running mini-batch EM with batch size 1024 and $\eta$ changing linearly from 0.1 to 0.05; full-batch EM is then used to finetune the parameters.

---

**Algorithm 4** Compile the PGM representation of a HCLT into an equivalent PC

---

1: **Input:** A PGM representation of a HCLT $\mathcal{G}$ (e.g., Fig. 3(c)); hyperparameter $M$
2: **Output:** A smooth and structured-decomposable PC $p$ equivalent to $\mathcal{G}$
3: **Initialize:** cache $\leftarrow$ dict() a dictionary storing intermediate PC units
4: **Sub-routines:** **PC_leaf**$(X_i)$ returns a PC input unit of variable $X_i$; **PC_prod**$(\{n_i\}_{i=1}^m)$ (resp. **PC_sum**$(\{n_i\}_{i=1}^m)$) returns a product (resp. sum) unit over child nodes $\{n_i\}_{i=1}^m$.
5: **foreach** node $g$ traversed in postorder (bottom-up) of $\mathcal{G}$ **do**
6:     **if** var$(g) \in \mathbf{X}$ **then** cache$[g] \leftarrow \big[$**PC_leaf**$\big(\text{var}(g)\big)$ for $i = 1 : M\big]$
7:     **else** # That is, var$(g) \in \mathbf{Z}$
8:        chs_cache $\leftarrow \big[$cache$[c]$ for $c$ in children$(g)\big]$ # children$(g)$ is the set of children of $g$
9:        prod_nodes $\leftarrow \big[$**PC_prod**$\big(\big[$nodes$[i]$ for nodes in chs_cache$\big]\big)$ for $i = 1 : M\big]$
10:        cache$[g] \leftarrow \big[$**PC_sum**$($prod_nodes$)$ for $i = 1 : M\big]$
11: **return** cache$[\text{root}(\mathcal{G})][0]$

---

## B.2 GENERATING PCs FOLLOWING THE HCLT STRUCTURE

After generating the PGM representation of a HCLT model, we are now left with the final step of compiling the PGM representation of the model into an equivalent PC. Recall that we define the latent variables $\{Z_i\}_{i=1}^4$ as categorical variables with $M$ categories, where $M$ is a hyperparameter. As demonstrated in Alg. 4, we incrementally compile every PGM node into an equivalent PC unit though a bottom-up traverse (line 5) of the PGM. Specifically, leaf PGM nodes corresponding to observed variables $X_i$ are compiled into PC input units of $X_i$ (line 6), and inner PGM nodes corresponding to latent variables are compiled by taking products and sums (implemented by product and sum units) of its child nodes' PC units (lines 8-10). Leaf units generated by **PC_leaf**$(X)$ can be any simple univariate distribution of $X$. We used categorical leaf units in our HCLT experiments. Fig. 3(d) demonstrates the result PC after running Alg. 4 with the PGM in Fig. 3(c) and $M = 2$.

## B.3 IMPLEMENTATION DETAILS OF THE PC LEARNING ALGORITHM

We adopted the EM parameter learning algorithm introduced in Choi et al. (2021), which computes the EM update targets using *expected flows*. Following Liu & Van den Broeck (2021), we use a hybrid EM algorithm, which uses mini-batch EM updates to initiate the training process, and switch to full-batch EM updates afterwards.

• Mini-batch EM: denote $\boldsymbol{\theta}^{(\text{EM})}$ as the EM update target computed with a mini-batch of samples. An update with step-size $\eta$ is: $\boldsymbol{\theta}^{(k+1)} \leftarrow (1 - \eta)\boldsymbol{\theta}^{(k)} + \eta\boldsymbol{\theta}^{(\text{EM})}$.

• Full-batch EM: denote $\boldsymbol{\theta}^{(\text{EM})}$ as the EM update target computed with the whole dataset. Full-batch EM updates the parameters with $\boldsymbol{\theta}^{(\text{EM})}$ at each iteration.

In our experiments, we trained the HCLTs with 100 mini-batch EM epochs and 20 full-batch EM epochs. During mini-batch EM updates, $\eta$ was annealed linearly from 0.15 to 0.05.

## B.4 DETAILS OF THE COMPRESSION/DECOMPRESSION EXPERIMENT

**Hardware specifications** All experiments are performed on a server with 72 CPUs, 512G Memory, and 2 TITAN RTX GPUs. In all experiments, we only use a single GPU on the server.

**IDF** We ran all experiments with the code in the GitHub repo provided by the authors. We adopted an IDF model with the following hyperparameters: 8 flow layers per level; 2 levels; densenets with depth 6 and 512 channels; base learning rate 0.001; learning rate decay 0.999. The algorithm adopts an CPU-based entropy coder rANS. For (de)compression, we used the following script: `https://github.com/jornpeters/integer_discrete_flows/blob/master/experiment_coding.py`.

**BitSwap** We trained all models using the following author-provided script: `https://github.com/fhkingma/bitswap/blob/master/model/mnist_train.py`. The al-

gorithm adopts an CPU-based entropy coder rANS. And we used the following code for (de)compression: `https://github.com/fhkingma/bitswap/blob/master/mnist_compress.py`.

**BB-ANS**  All experiments were performed using the following official code: `https://github.com/bits-back/bits-back`.

### B.5  DETAILS OF THE PC+IDF MODEL

The adopted IDF architecture follows the original paper (Hoogeboom et al., 2019). For the PCs, we adopted EiNets (Peharz et al., 2020a) with hyperparameters $K = 12$ and $R = 4$. Instead of using random binary trees to define the model architecture, we used binary trees where "closer" latent variables in $z$ will be put closer in the binary tree.

Parameter learning was performed by the following steps. First, compute the average log-likelihood over a mini-batch of samples. The negative average log-likelihood is the loss we use. Second, compute the gradients w.r.t. all model parameters by backpropagating the loss. Finally, update the IDF and PCs using the gradients individually: for IDF, following Hoogeboom et al. (2019), the Adamax optimizer was used; for PCs, following Peharz et al. (2020a), we use the gradients to compute the EM target of the parameters and performed mini-batch EM updates.

