# OpenReview forum: "Lossless Compression with Probabilistic Circuits"
_ICLR.cc/2022/Conference — ICLR 2022 Spotlight_

### Official Review · Reviewer_WvLE · 2021-11-02

**Correctness:** 3
**Technical Novelty And Significance:** 3
**Empirical Novelty And Significance:** 3
**Recommendation:** 6
**Confidence:** 2

**Main Review:**

Strengths:
The contributions of the paper are: (i) reducing the complexity with respect to D from linear to sub-linear and (ii) integration with existing neural compressors. The authors use Hidden Chow-Liu Tree (HCLT) , a PC model initially proposed for simple density estimation tasks containing binary features, to scale up to achieve state-of-the-art performance on various image datasets.  Furthermore since HCLTs cannot be easily vectorized, the authors implemented customized GPU kernels for parameter learning and marginal query computationbased on Juice.jl, an open-source Julia package. This way the authors were able to get significant improvement in the performance.

Weaknesses:
 The baselines BitSwap and IDF, which are definitely NOT the state of the art in compression efficiency and compute time. The comparison that is missing is with respect to 'Improving Lossless Compression Rates via Monte Carlo Bits-Back Coding' (Ruan, Ullrich, Severo, et al.). In that paper, the authors claim to compress all 10,000 MNIST images in less than 100s (Figure 7). Comparison with BitSwap (and possibly IDF) should be done with caution. In BitSwap, the entropy coder runs on CPU while the model can be run on GPU or CPU. It's not clear what the experimental setup was here for BitSwap, PC, and IDF experiments. Exactly what setup the authors claim SOTA should also be clarified (e.g. model on GPU, compression on CPU).


Comments regarding writing
Page 4:
- Not clear what happens when 2 different input units have the same variable. This is exactly the case of Figure 1, where there are 2 input nodes for each of X1, X2, and X3.

Page 5:
- If the inputs are descendants of the same child unit, then doesn't this imply the opposite? That is, you definitely need to multiply them based on equation 2? It would be easier if the authors followed along with a concise and concrete example.
- The example in Figure 1 does not help, as the distance from n1 and n2 to the root node is too short for the reader to properly follow along.



**Summary Of The Paper:**

The paper showcases an application of Probabilistic Circuits to lossless compression, and achieves competitive compression performance to state of the art method. The PCs output a marginal distribution over the data, which the authors then use to compress with arithmetic coding or other methods. This has several advantages, one being that bits-back coding is not needed, enabling single-sample compression.



**Summary Of The Review:**

Overall the work is interesting and timely and I am leaning towards an accept if the authors can convincingly address the points above.

---

> ### Author Response · Authors · 2021-11-22
> **Response to Reviewer WvLE**
>
> We thank the reviewer for appreciating the novelty of our method.
>
> > Comparison with McBits
>
> There seems to be some misunderstanding regarding the compression time of MNIST. In the McBits paper, the ~100s compression time reported in Figure 7 is for *binarized* MNIST, which is 8 times smaller than the raw MNIST, where we reported our results.
>
> Since the McBits authors did not report results on raw MNIST, we tried to run their code to report the numbers. Unfortunately, their coder found on GitHub seems to have some non-trivial bugs that we could not fix; thus we were not able to run encoding/decoding with the raw MNIST model (en- and decoding of the binarized MNIST model worked very well). We have added theoretical bpds of McBits in Table 3 and will seek help to get the coder running and will add the compression bitrates to Tables 2 and 3. The proposed compressor is still SoTA on MNIST and EMNIST.
>
> > Comparison with BitSwap and IDF. … Exactly what setup the authors claim SOTA should also be clarified (e.g. model on GPU, compression on CPU)
>
> For both the proposed compressor and the adopted baselines, the entropy coder (rANS) is implemented in CPU. This is stated in footnote #7 and Appx. B.4 in the revised paper. Although this could be potentially slower than a customized GPU implementation, we note that the proposed PC-based compression algorithm also uses a CPU implementation of rANS. In addition, the entropy coder takes almost negligible time compared to the computation time required by the entropy model (neural networks and PCs). Therefore, in our humble opinion, the timing comparison with BitSwap and IDF can be considered fair.
>
> > Not clear what happens when 2 different input units have the same variable. This is exactly the case of Figure 1, where there are 2 input nodes for each of X1, X2, and X3.
>
> In this case, the two input units represent two (different) distributions on the variable X. For example, they can represent two Gaussian distributions, and if we add a sum unit on top of them, the distribution represented by the sum unit is a mixture of two Gaussians.
>
> > If the inputs are descendants of the same child unit, then doesn't this imply the opposite? That is, you definitely need to multiply them based on equation 2? It would be easier if the authors followed along with a concise and concrete example.
>
> The reason why we do not multiply $p_{n_1}(x)$ and $p_{n_2}(x)$ is that for each decomposable product node $m$, only distributions defined on disjoint variables shall be multiplied. Since $n_1$ and $n_2$ have the same variable scope, their distributions will not be multiplied by any product node. We have added this more intuitive explanation in the revised paper.
>
> > The example in Figure 1 does not help, as the distance from n1 and n2 to the root node is too short for the reader to properly follow along.
>
> In the updated paper, we have rewritten Sec. 3.1 and 3.2 to make the proposed algorithm easier to understand. Specifically, in Sec. 3.2, instead of directly delving into technical details (e.g., why do we need the top-down probabilities), we intuitively demonstrate (i) why we can compute the marginals in $F(x)$ without the need to evaluate all PC units and (ii) what types of PC units can be safely pruned for evaluation. We hope the new Sec. 3.2 can provide readers with a better high-level idea of the algorithm, and makes the technical parts easier to understand.

---

### Official Review · Reviewer_H7Ew · 2021-11-03

**Correctness:** 3
**Technical Novelty And Significance:** 2
**Empirical Novelty And Significance:** 3
**Recommendation:** 8
**Confidence:** 4

**Main Review:**

A strength of this paper is its clever use of PCs in compression. First, it highlights the importance of tractability in lossless compression, which is an unexplored perspective in related works. Then, PCs are presented as a class of models with tractable inference, while still being expressive enough. This balance between expressiveness and computational cost is key this manuscript's main contribution.

It is worth clarifying the significance of the technique for improving marginal inference from O(D.|p|) to O(log(D).|p|). This algorithm has relevant similarities with Partial Propagation described in [1]. Thus, a comment on differences here could highlight the novelty in Algorithm 2.

Section 4 provides a practical way for scaling up the use of PCs in compression. However, it is not clear from the manuscript the contributions made in the scaling up process. That is, if there was any new required technique developed for applying Hidden Chow-Liu Tree to the specific problem of compression.

Experimental results are encouraging and convincing.


Minor comments:
* There is a typo with "p(i.e.," in Section 2
* Theorem 1 significance is not clear from the manuscript and it seems incremental at a first glance. While there is no questioning of its practical implications, a formal treatment could highlight its broader applications.


[1] C.J. Butz, J.S. Oliveira, A. dos Santos, A.L. Teixeira, P. Poupart, A. Kalra, An Empirical Study of Methods for SPN Learning and Inference, Ninth International Conference on Probabilistic Graphical Models (PGM), 49--60, 2018.

**Summary Of The Paper:**

The manuscript addresses the question of using deep generative models in lossless compression. As highlighted by the authors, the main issue here is related to the cost of computing probabilistic queries or, in some cases such as GANs, the inability to compute them.
Thus, this paper suggest using Probabilistic Circuits (PCs) for lossless compression. PCs allow for tractable probabilistic inference, which enables efficient compression.

Experimental results are favourable in two ways. First, PCs are faster for compression, achieving results from 5 to 20 times faster than competitor neural networks. Second, PCs achieve competitive compression rates on various datasets.

**Summary Of The Review:**

The manuscript provides a class of tractable models for lossless data compression. These models are well defined and algorithms are shown. Parts of the work need clarifying for novelty and significance. However, experiment results with faster and competitive compression rates are very convincing.

---

> ### Author Response · Authors · 2021-11-22
> **Response to Reviewer H7Ew**
>
> We thank the reviewer for praising this paper for providing a clever use of PCs in compression.
>
> > It is worth clarifying the significance of the technique for improving marginal inference from O(D.|p|) to O(log(D).|p|).
>
> Thank you for pointing out this related work. The proposed algorithm is related to the Partial Propagation (PP) algorithm proposed by Butz et, al. (2018) in the sense that they both seek to compute certain queries by only evaluating a subset of PC nodes. The key difference between the proposed algorithm and PP is that we exploited the structure of the queries needed for compression, and as a result, we identified *more* PC nodes that do not need to be evaluated. Specifically, as discussed in paragraph 7 of Sec. 3.2, while PC only prunes away nodes in Group #2, the proposed algorithm also prunes away nodes in Groups #1 and #3. This is the key to *guarantee* O(log(D) |p|) time complexity.
>
> > However, it is not clear from the manuscript the contributions made in the scaling up process
>
> The scaling-up process is mainly about the improved implementation of the learning and inference algorithms that better utilize GPU capacity. Specifically, since the Hidden Chow-Liu Tree has an irregular PC structure, the model cannot be easily vectorized as done in [1]. Therefore, to improve the learning/inference efficiency, we wrote customized GPU kernels based on the Julia PC package Juice.jl. In addition, we found the following training strategy leads to a better test set performance: first train the PC with mini-batch EM, and then finetune the PC with full-batch EM. All the above information can be found in paragraph 2 of Sec. 4.2 and Appx. B.3.
>
> > Theorem 1 significance is not clear from the manuscript and it seems incremental at a first glance.
>
> In the revised paper, we have re-written Sec. 3.2 to better reflect the significance of Theorem 1. Specifically, we used Fig. 2 to demonstrate how we exploit the structure of the marginal queries $F(x)$ to compute them by evaluating the minimum number of PC units. We also added a detailed discussion in paragraph 7 of Sec. 3.2 to compare with the Partial Propagation algorithm proposed by Butz et, al. (2018) --- we exploited the structure of $F(x)$ and pruned away more PC units, which is the key to guaranteeing the efficiency of the proposed algorithm.
>
> > There is a typo with "p(i.e.," in Section 2
>
> Thank you for pointing out the typo. We have fixed it in the revised paper.
>
> [1] Peharz, Robert, et al. "Einsum networks: Fast and scalable learning of tractable probabilistic circuits." International Conference on Machine Learning. PMLR, 2020.

---

### Official Review · Reviewer_mSD5 · 2021-11-04

**Correctness:** 3
**Technical Novelty And Significance:** 3
**Empirical Novelty And Significance:** 3
**Recommendation:** 5
**Confidence:** 3

**Main Review:**

Strengths:
This paper provides a new probabilistic modelling framework named PC for lossless compression. This framework is new in AI lossless compression community and achieves desired compression ratio and compression bandwidth in MNIST. By incorporating PC with explicit generative models in which the prior can be replaced with PC, the model achieves SOTA compression ratio in real-color images.

Weaknesses:
This work is somewhat hard to follow in the following aspects:
1.	In the abstract, the authors highlighted the drawback of VAEs such that “bits-back coding brings poor single-sample compression rates”. However, the proposed PC seems have no relevant with bits-back coding. I wonder what is the advantage of PC over VAEs in terms of “bits-back coding”?
2.	The authors claim that the complexity of PC is O(log D |p|) where |p| is number of neural network units. But I am not able to figure out how to incorporate neural networks with PC in O(log D |p|). In particular, consider flow models with PC, it just seems to replace the prior with PC, to what follows, the complexity seems to be O(log D + |p|). Moreover, for auto-regressive models, it seems that PC cannot reduce the complexity to O(log D |p|) but remains O(log D |p|). More discussions on how to incorporate generative models with PC with O(log D |p|) complexity is encouraged.
3.	The PC framework is hard to follow. In Definition 2-4, it is better to give definitions on “scope of variable”, and examples on “Smoothness”, “Decomposability” and “SD”. For Fig. 1, detailed probability corresponding to the figure is recommended.
4.	For Sec. 5, the compression bandwidth of PC+IDF compared with IDF is recommended.


**Summary Of The Paper:**

This paper proposes a new probabilistic model named PC for efficient encoding and decoding. The authors claim that PC greatly reduces the time complexity for inference. Experiments show that PC achieves 5-20x faster than SOTA neural compression algorithms, and performs SOTA results on MNIST datasets.

**Summary Of The Review:**

The idea of PC for lossless compression is new and interesting. However, following the main contribution of PC and the main idea of PC is somewhat difficult. Giving simple examples on PC for clarification is recommended.

---

> ### Author Response · Authors · 2021-11-22
> **Response to Reviewer mSD5**
>
> We thank the reviewer for their helpful comments and feedback, and to appreciate the novelty and significance of our work.
>
> > The PC framework is hard to follow
>
> As mentioned in the general response to all reviewers, we faced the difficulty of bridging two highly unrelated and nontrivial communities --- neural compression and probabilistic circuits. In the revised paper, we made significant efforts to improve the writing of the method section to make it easier for researchers unfamiliar with PC to understand. Specifically, we made the following changes to the paper:
>
> - While introducing the syntax and semantics of PCs (Sec. 3.1), we added a concrete example PC (Fig. 1) to explain related concepts, including the definition of scope, decomposability, structured-decomposability, and how to compute arbitrary marginal queries given a smooth and decomposable PC.
>
> - The PC-based (de)compression algorithm is better elaborated. We rewrote Sec. 3.2 to highlight the following key points: (i) the high-level workflow of the proposed algorithm is shown in Fig. 2 and elaborated in the first paragraph of Sec. 3.2; (ii) as a backbone procedure of the en- and decoder, we intuitively demonstrate why the $D$ marginals can be computed efficiently and how that is done (Alg. 1). With more intuition added in the revised paper, we hope the method can be understood more easily.
>
> - Specifically, we use Fig. 2 to illustrate why we need to compute the $D$ marginal probabilities in $F(x)$ and how these probabilities are used by a streaming code for en- and decoding.
>
> - By drawing an analogy between PCs and neural networks, we intuitively show why we can compute each term in $F(x)$ by partially evaluating the model. We further demonstrate what types of PC units do not need to be evaluated in a more intuitive way.
>
> > I wonder what is the advantage of PC over VAEs in terms of “bits-back coding”?
>
> “Bits-back coding” is an elaborate coding technique for lossless compression in VAEs that realizes an expected code length that equals the evidence lower bound; the disadvantage is that this technique is only space-efficient when encoding many images in sequence. We need to refer to the literature (e.g., Townsend 2019) for details. PCs do not require bits-back coding to obtain close-to-optimal (i.e., the limit specified by the source coding theorem) compression rate, thereby being both faster and efficient in encoding individual images.
>
> > But I am not able to figure out how to incorporate neural networks with PC in O(log D |p|)
>
> The O(log D |p|) time complexity holds only for the PC model, not for the neural network model. Specifically, for the PC+IDF model used in the paper, if the PC has |p1| neurons and the neural network model of IDF has |p2| neurons, then the overall time complexity for both encoding and decoding is O(log(D) * |p1| + |p2|). We clarified this in the last paragraph of Sec. 5.
>
> > compression bandwidth of PC+IDF compared with IDF is recommended
>
> We did not implement the en- and decoder for PC+IDF because our PC-based en- and decoder was implemented in Julia with customized GPU kernels while IDF is implemented via Python+PyTorch. We highlight that the main goal of these experiments is to show that PCs can be hybridized with Flow models such as IDF to significantly improve their likelihoods, and it would be valuable to explore such integrations further in future work.

---

> > ### Comment · Reviewer_mSD5 · 2021-11-29
> > **Post-rebuttal**
> >
> > I am satisfied with detailed illustration of PC, which is much clearer to follow.
> > However, there are some suggestions to make the paper better:
> > 1.	For coding with VAE, I hope to find a coding algorithm that avoids bits-back coding. Unfortunately, it seems that PC could not solve this issue as decoding with p(z|x) is unlikely to be avoided with PC. In fact, PC performs much faster than auto-regressive models, which can be highlighted.
> > 2.	For PC+IDF, the authors admit that the coding time is hard to summarize due to the coders, and the complexity is similar compared with IDF++. But on the other hand, I suggest just summarize the inference time between IDF++ and PC+IDF, and show the time consumption by replacing the mix-logistic prior to PC.
> > 3.	It seems that PC model is a non-neuron model, thus I doubt the performance on complex images. Results on complex images like CIFAR10, ImageNet32, ImageNet64 with ONLY PC model are recommended.

---

> > > ### Author Response · Authors · 2021-11-29
> > > **Response**
> > >
> > > Thank you for recognizing the efficiency of the PC-based compressor compared to autoregressive models. We will add an inference time comparison between IDF and PC+IDF in the next version of the paper.

---

### Official Review · Reviewer_T76k · 2021-11-04

**Correctness:** 2
**Technical Novelty And Significance:** 3
**Empirical Novelty And Significance:** 2
**Recommendation:** 6
**Confidence:** 3

**Main Review:**

Overall, I felt that whilst this was a reasonably well written paper highlighting the interesting concept of PCs, that I was not previously aware of, the methodological contribution wasn't clear enough and that the experiments were presented in a way which was potentially misleading.

My main issues with the methodological contribution (specifically section 3.2), are that I felt it was relatively difficult to understand (i.e. the presentation could have been clearer), and that as a result I don't know how significant the contribution is. On a high level, it seemed like a couple of observations were made:

(a) For many models, it is important to decode/encode data in an order which 'fits' the model, leading to computational efficiency.
(b) Work sharing between encode/decode steps is necessary for efficiency.

To someone, such as myself, who is an expert on neural lossless compression, these are both obvious points. The O(log D |p|) computational efficiency didn't seem particularly radical to me either. However, I'm unfamiliar with the PCs literature, so maybe these are significant observations to that community, and it's possible that I missed something important, so please clarify.

Before I discuss the experiments, another significant issue with the overall framing was the statement in the second paragraph of the introduction that the methods presented here are 'tractible', with the implication being that other neural compression methods are not (in some sense). I understand where this came from, since tractibility is a key property in the PCs literature, but I still felt this could be misleading, since neural compression and decompression with existing methods, whilst they may have different performance characteristics to PCs, certainly _are_ tractible.

I had two main issues with the experiments. The first relates to the presentation of the timings of the method. I think it's easy to be misled, and to mislead, with timings in machine learning (and the emerging field of neural lossless compression in particular), and my issue with timings in this work is that the paper is presented as though a significant breakthrough has been made in terms of runtime, but the PC method is only compared to slow implementations of existing methods, which were not optimized for speed. One example of a faster implementation of a neural compression method is BB-ANS with a small VAE (from Townsend et al., 2019), implemented at https://github.com/j-towns/craystack. Running the example code there on a CPU, the compression/decompression for the binarized MNIST test set (a slightly easier task than raw MNIST) is 3.26s/2.82s, an order of magnitude faster than the PC timings in the paper. For extra context, on the same machine gzip takes 0.22s to compress and 0.06s to decompress the raw MNIST test set, so in my opinion neither the Craystack BB-ANS implementation nor the 15s encode and 44s decode of the PCs implementation should be considered fast. This context should have been made clearer, rather than trying to imply that a genuine breakthrough had been made in runtime.

My other serious issue with the presentation of the experiments were the vague claims of "state-of-the-art performance on natural image datasets", in the abstract and at the very end of section 2, discussed in more detail in section 5. It's not at all clear on what task the authors are claiming state of the art performance. At the end of sec 5, the suggestive statement is made that compression/decompression of natural images "can be done easily" with the implemented method. Have the authors actually implemented this? If compression wasn't implemented then this should be stated and the authors should explain why not, if it was implemented this should be clearly stated. Unfortunately, even if compression was implemented, a claim of state of the art performance would still be incorrect, because the bitrates achieved in a recent paper by Zhang et al. (https://arxiv.org/abs/2109.02639) are significantly better.

Some more minor points and suggestions:
 - At the beginning of sec 3.2, why does \pi need to be 'random'? Don't you just mean to say that \pi is "some ordering" (i.e. no need to suggest that it is a random variable).
 - In section 4.1 I think it's best to consistently use the word 'latent' rather than 'hidden', since this is the standard terminology in the deep generative modelling community.
 - It might be a good idea to combine the different related work sections in one place.
 - In 'related work' near the bottom of page 8, there's a slight grammatical error: replace "grow PC structures to fit better the data" -> "grow PC structures to better fit the data".

**Summary Of The Paper:**

Probabilistic circuits are a formalism, developed relatively recently, for describing multivariate probability distributions. PCs are represented using directed acyclic graphs (DAGs), with an operational semantics, and it is relatively straightforward to deduce which operations (marginalization, maximisation, estimation of moments, etc.) are tractible based on the structure of the DAG and the locations of input variables.

This paper studies the application of PCs to lossless compression, identifying the specific type of marginalisation which is necessary for auto-regressive, arithmetic coding-style compression, and the necessary conditions a PC must satisfy in order for this marginalisation to be tractible. In section 3, a procedure is described for particularly efficient marginalisation by sharing computation.

Experiments show that the approach is competitive in terms of compression rate with other recent 'neural compression' works, and results are presented showing that PCs can be much faster than some existing methods, although I have serious concerns about these (see below).

**Summary Of The Review:**

I felt that whilst this was a reasonably well written paper highlighting the interesting concept of PCs, which I was not previously aware of, the methodological contribution wasn't clear enough and that the experiments were presented in a way which was potentially misleading.

EDIT: Score changed to 6, see comment below.

---

> ### Author Response · Authors · 2021-11-22
> **Response to Reviewer T76k**
>
> We thank the reviewer for their helpful comments and suggestions.
>
> > this was a reasonably well written paper highlighting the interesting concept of PCs, that I was not previously aware of
>
> We thank the reviewer for recognizing our work as an initial effort to bridge two highly diverse communities. In particular, we show that PCs are suitable for many lossless image compression tasks while being faster than other algorithms with SoTA bitrates.
>
> > My main issues with the methodological contribution (specifically section 3.2), are that I felt it was relatively difficult to understand (i.e. the presentation could have been clearer), and that as a result I don't know how significant the contribution is.
>
> As mentioned in the general response, while writing this paper, we faced the dilemma of balancing between (i) making the method easy to understand by researchers not familiar with PCs and (ii) describing non-trivial technical details that ensure the algorithm runs in $O(|p| \cdot \log D)$ time.
>
> We want to highlight that the main contributions of this paper are (i) using PCs, a new ‘niche’ class of deep generative models, for lossless compression, (ii) demonstrating that we can learn PCs that perform on par or even better than many traditional deep generative models, and (iii) proposing a fast (de)compression algorithm.
>
> During the rebuttal, we made the following changes to the paper to better introduce our (de)compression algorithm to researchers unfamiliar with PCs:
>
> - A better background section on PCs. We added a concrete example in Sec. 3.1 to explain related concepts, including variable scope, decomposability, and how to compute arbitrary marginal probability given a smooth and decomposable PC.
>
> - A high-level overview of the proposed PC-based en- and decoder. We have added Figure 2 to demonstrate the workflow of the proposed algorithm, emphasizing why we need to compute the marginal probabilities $F(x)$ and how they are used by a streaming code to en- and decode the data.
>
> - A better description of how the $D$ marginals in $F(x)$ can be computed efficiently. In paragraphs 5-7 of Sec. 3.2, we made an analogy between PCs and neural networks to intuitively demonstrate why we can compute each marginal by only evaluating a small fraction of PC units. We then formally described what type of PC units do not need to be evaluated.
>
> > presentation of the timings of the method could be misleading.
>
> We totally agree that the timing results should be reported in a more precise way. Specifically, in Table 2, we intended to compare the en- and decoding time of PCs with other compression algorithms that achieve near SoTA compression rates. Results in Table 2 demonstrate that PCs are much faster than other neural compression algorithms with near SoTA bitrates.
>
> To better represent the whole spectrum of bitrate vs. elapsed time, we have added a paragraph in Sec. 3.3 to discuss the tradeoff between compression rates and speed, including the discussion of Townsend et al. (2019) and other related papers.
>
> To avoid further confusion, we have rephrased the abstract and introduction to emphasize we are only faster than “compression algorithms that achieved similar bitrates”.
>
> > vague claims of "state-of-the-art performance on natural image datasets"
>
> We thank the reviewer for the suggestion of making the claims more precise. For the experiments on natural image datasets, we wanted to demonstrate that PCs can be naturally integrated with existing Flow-based compression algorithms to achieve better performance. In the revised paper, we have made this claim precise in the abstract and the last paragraph of Sec. 2. We also added a discussion on concurrent work (submitted to ArXiv one month before the ICLR deadline) [Zhang et al.], which achieved SoTA performance on datasets such as CIFAR-10.
>
> > Have the authors actually implemented this? If compression wasn't implemented then this should be stated and the authors should explain why not, if it was implemented this should be clearly stated.
>
> The compression algorithm for PC+IDF is not implemented mainly due to implementation issues. Specifically, IDF is implemented in Python+Pytorch, while the PC compressor (see the submitted code) is implemented in Julia, with customized GPU kernels.
>
> > statement in the second paragraph of the introduction that the methods presented here are 'tractible', with the implication being that other neural compression methods are not (in some sense) could be misleading.
>
> Thank for the suggestion. Indeed the previous phrasing could be misleading. We have rewritten the second paragraph of the introduction to better position our work among other compression algorithms.

---

> > ### Author Response · Authors · 2021-11-22
> > **Response to Reviewer T76k (part 2)**
> >
> > > At the beginning of sec 3.2, why does \pi need to be 'random'? Don't you just mean to say that \pi is "some ordering" (i.e. no need to suggest that it is a random variable).  ….  In 'related work' near the bottom of page 8, there's a slight grammatical error: replace "grow PC structures to fit better the data" -> "grow PC structures to better fit the data".
> >
> > Thank you for pointing out both typos. We have fixed them in the revised paper.
> >
> > > In section 4.1 I think it's best to consistently use the word 'latent' rather than 'hidden', since this is the standard terminology in the deep generative modelling community.
> >
> > Thank you for the suggestion. We have switched to using the word “latent” in the revised paper.

---

> > > ### Comment · Reviewer_T76k · 2021-11-29
> > > **Thanks for your response and for the changes.**
> > >
> > > It seems that overall, significant improvements have been made to the paper. I've changed my score to a weak accept. I would greatly appreciate it if the authors could explicitly state _in the paper_ that they have not implemented compression with the PC+IDF model. For example, by changing the sentence beginning "Compression and decompression with the PC+IDF model can be done easily..." to "Although we leave the implementation to future work, compression and decompression with the PC+IDF model should be straightforward since...".
> > >
> > > I'm still slightly concerned about the way PCs are presented as being strictly + significantly faster than other "suboptimal" DGM-based appoaches. Surely in practice this would depend on things like the size of the DGM, and presumably the implementation and the hardware used. In a fully optimized implementation, is it not possible, for example, that a small VAE (with say one or two convolutional layers) would outperform a PC on speed, because of better vectorization and parallelization? It's a shame that Townsend (2019) doesn't provide timings for full MNIST, since the compression rate of the small VAE (1.41bpd) is actually not too far from the PC rate.

---

> > > > ### Author Response · Authors · 2021-11-29
> > > > **Response**
> > > >
> > > > Thank you for the helpful comments. We will rephrase the corresponding sentences according to your suggestions and discuss runtime vs. bitrate in more detail to better position the proposed compressor.

---

### Author Response · Authors · 2021-11-22
**A general response to all reviewers**

We thank all reviewers for their helpful comments and feedback. All reviewers recognize that our work proposes a novel perspective to lossless compression using Probabilistic Circuits, a class of deep probabilistic models previously unexplored in related fields. The two main concerns were that (i) the method is relatively hard to understand for researchers not familiar with PCs and (ii) some experiment results need to be better explained to avoid confusion. In the revised paper, we made significant efforts to address both concerns:

- To make the paper more comprehensible to researchers unfamiliar with PCs, we rewrote Sec. 3 in the revised paper to provide more intuitive elaborations and high-level ideas. Specifically, in Sec. 3.1, we added an example PC that is used to explain introduced concepts, including variable scope, decomposability, structured-decomposability, and how to compute marginal queries. We also revised Sec. 3.2 to better explain the PC-based compressor/decompressor. Specifically, we first provide a high-level overview of the en- and decoding process. Only later, we delve into explaining the backbone algorithm that computes the required marginal probabilities.

- We made our main experimental claims and metrics more precise. Reviewer T76k suggests that we should be cautious claiming that the PC-based compression algorithm is fast, since there exist faster compression algorithms, though at the cost of lower bitrates. In the paper, we were only comparing the timing with compression algorithms that achieve close-to-SoTA bitrates. In the revised manuscript, we have explained our metric clearly to avoid further confusion. We also added a discussion showing the existence of faster compression algorithms with worse bitrates.

- We included additional baselines in our paper that the reviewers pointed out, including a very recent one. Our results still indicate competitive bitrates on MNIST, Fashion, and EMNIST.

Finally, we acknowledge that by drawing on deep contextual knowledge from two unrelated fields--neural data compression and probabilistic circuits--we challenge our readers. We hope that the reviewers generally agree that this kind of research has the potential to yield non-incremental new results. We hope the above changes can help people better appreciate the potential impact of PCs and other tractable probabilistic on the compression community.

---

### Decision · Program_Chairs · 2022-01-20

**Decision:**

Accept (Spotlight)

**Comment:**

The paper revisits lossless compression using deep architecture. In contrast to main stream approaches, it suggests to make use of probabilistic circuits, introducing a novel class of tractable lossless compression models. Overall, the reviews agree that this is an interesting direction and a novel approach. I fully agree. Actually, I like that the paper is not just saying well, we could use a probabilistic circuit for ensure tractability but also shows that there is still a benefit of different variable orderings for encoding and decoding. In any case, adding probabilistic circuits to the "compression family" is valuable and also paves the way to novel hybrid approaches, combining neural networks and probabilistic circuits. I have enjoyed reading the paper, reviews, and discussion.